# Knowledge, attitudes and practices towards COVID-19 among healthcare workers: A cross-sectional survey from Kiambu County, Kenya

Prabhjot Kaur Juttla[1]*, Moses Ndiritu[2], Ferdinand Milliano[2], Alfred Owino Odongo[3], Magoma Mwancha-Kwasa[2]

1 Faculty of Health Sciences, School of Medicine, University of Nairobi, Nairobi, Kenya, 2 Department of Health, County Government of Kiambu, Kiambu, Kenya, 3 School of Public Health, Mount Kenya University, Thika, Kenya

* pkjuttla13@gmail.com

## Abstract

### Background

The knowledge possessed by healthcare workers (HCWs), along with their attitudes and practices play a vital role in effectively managing a pandemic. This is crucial considering that HCWs are exposed to great risk at the forefront of such crises. We aimed to describe the knowledge, attitude, and practices (KAP) of HCWs during the COVID-19 pandemic in Kiambu county, Kenya.

### Methods

A cross-sectional study using a structured questionnaire was conducted from 11th March 2021 to 12th August 2021. Bloom's cutoff points were used to determine KAP scores (>80%: good, 60–79%: medium and <60% poor). Multivariable ordinal logistic regression analyses were conducted, calculating adjusted odds ratios (AOR) at a 95% confidence interval. Spearman's rank correlations were used to examine the relationship between KAP scores.

### Results

438 HCWs participated in the study, majority of whom were female (64.5%), had obtained a diploma (59.6%) and were informed through government websites (78.6%). 43.0% had good knowledge, 17.5% good attitudes, and 68.4% good practice. 23.0% had medium knowledge, 35.6% medium attitude, 15.7% medium practice, while 34.0% had poor knowledge, 46.9% poor attitude and 15.9% poor practice. Only 68.9% of the caregivers correctly recognized the county's COVID-19 isolation centre and only 7.9% chose the two correct documents for the entry of data for a suspected COVID-19 case. Furthermore, the general attitude towards their own personal safety and their training regarding PPEs (17.8% and 23.8% strongly agreed with the contrary) were less compelling. There was a significant positive association between obtaining information from international government sites [AOR: 1.382 (1.058–1.807); p = 0.0178)] and good knowledge. Referring to local government sites

**Data Availability Statement:** The data underlying this research can be found in the S1 File.

**Funding:** The author(s) received no specific funding for this work.

**Competing interests:** The authors have declared that no competing interests exist.

for information regarding COVID-19 produced better attitudes [AOR: 1.710 (1.194–2.465); p = 0.0036] and produced almost twice the odds of having better practice [AOR: 1.800 (1.181–2.681); p = 0.0048]. There was a significant correlation between knowledge and practice ($r$ = -0.330, p = 2.766×10$^{-11}$), and knowledge and attitude ($r$ = -0.154, p = 6.538×10$^{-3}$).

## Conclusion

This study emphasizes the substantial impact that governing bodies have on shaping favorable KAP. As a result, it's crucial for local government platforms to prioritize the dissemination of up-to-date information that aligns with international standards. This information should be tailored to the specific region, focusing on addressing deficiencies in healthcare practices and patient management. The identification of a significant number of HCWs lacking confidence in managing COVID-19 patients and feeling unprotected underscores a clear need for improvement in their understanding and implementation of preventive measures. This gap can be bridged by adequately equipping HCWs with locally manufactured PPEs. This aspect is crucial for pandemic preparedness, and we further advocate for the creation of a locally produced repository of medical equipment. These actions are pivotal in improving future crisis management capabilities.

## Introduction

The global COVID-19 pandemic was an unprecedented risk to public health globally, prompting the World Health Organization (WHO) to declare it a public health emergency in early 2020 [1, 2]. Governments worldwide implemented non-pharmaceutical interventions, including travel bans, economic shutdowns, and social isolation, to stop the virus's spread [2, 3]. Despite these initiatives, two million COVID-19-related deaths and over 100 million confirmed cases of infection had been reported by early 2021—roughly a year after the pandemic started [4].

At the frontline of this pandemic were healthcare workers (HCWs). This came at a hefty price given that the WHO reported that 14% of COVID-19 patients were HCWs (even though they comprise less than 3% of the population in most countries) [2]. This high morbidity has been partially attributed to a lack of knowledge and misunderstandings among HCWs contributing to delayed diagnosis, spread of disease and subpar infection control practice [5]. Such high morbidity rates may have also made HCWs reluctant to continue caring for patients, which would inevitably weaken response efforts [6].

Numerous studies have looked into HCWs' knowledge, attitudes, and practices (KAP) about COVID-19. According to a cross-sectional research conducted in Eastern Ethiopia, 61.5% of participants had effective COVID-19 preventative measures, 54.8% had a positive attitude, and 73.3% of participants had sufficient knowledge [7]. According to the aforementioned study, there was a negative relationship between practice and knowledge as well as between knowledge and attitude [7]. Research conducted in Nepal discovered that 78.9% of HCWs indicated suitable practice, 54.7% expressed favourable attitudes, and 76% reported sufficient understanding about COVID-19 [8]. This study conducted in Nepal found a positive correlation between KAP. Another study with majorly HCWs in Asian countries found that, despite their lack of understanding of COVID-19, they demonstrated a positive attitude

towards its prevention [9]. This study also indicated that HCWs' professions corresponded to low levels of knowledge; however, research among HCWs in Uganda revealed no discernible relationship between a HCW's cadre and their knowledge levels [10]. In the same Ugandan study, age and news media were linked to adequate knowledge, with 69% having good knowledge, 74% having beneficial practices, but only an abysmal 21% having a positive attitude. Another study that surveyed emergency medical service providers in Turkey discovered that the majority had unfavourable attitudes toward the usage of personal protective equipment (PPE) and lacked sufficient knowledge regarding proper disinfection protocols [11]. These investigations demonstrate the great range and distinctions in HCWs' KAPs concerning COVID-19. These figures also diverge from those of a worldwide systematic study that found that, in the early stages of the pandemic, at least 75% of healthcare workers had good KAP scores towards COVID-19 [12].

The facts and practices surrounding a novel public health emergency are ever-evolving based on new research. But rather than having easily pliable attitudes, people typically maintain stable attitudes [13]. In addition to this resistance, people seek information already established within their psyche to reinforce their original attitudes and actions [13]. New facts and information seldom address this phenomenon [14]. This phenomenon would not be expected to change during the COVID-19 pandemic. However, given the paramount role of HCWs in pandemic response, what they know, feel and do becomes crucial in the face of a public health emergency. This situation is further complicated by the "infodemic" that paralleled the COVID-19 pandemic [15], and by a supply chain crisis that affected the availability of PPEs for adhering to proper protocols [16]. To avert the higher rate of being infected among HCWs, equipping them with good knowledge and practice is imperative, especially in countries with already low health worker-to-population ratio [17]. Thus, the KAP of HCW towards COVID-19 were critical to the success of the overall COVID-19 response. To explore this hypothesis, we carried out a cross-sectional study aimed at describing HCWs' KAP regarding COVID-19 in Kiambu County, Kenya.

## Materials and methods

### Study design

This was a cross-sectional online and hardcopy survey conducted using a self-administered structured questionnaire targeting the healthcare workers of Kiambu County, Kenya. The tools were available from the 11th March 2021 to 12th August 2021 which coincided with the third COVID-19 wave in Kenya [18]. The questionnaire was adopted from a similar study in English, which is one of the national languages of Kenya. The online form was administered via Google Forms, which was accessible by clicking on a link. Authors disseminated the link via specific HCW groups on WhatsApp and a hard copy tool was also made available to HCWs in the various healthcare facilities within Kiambu county. The respondents above 18 years old and who were HCWs working in Kiambu county who wished to provide the information were requested to fill in the information.

### Study setting

Kiambu County (-1˚10'0.01" S 36˚49'59.99" E) is part of central Kenya. It neighbours Nairobi, Murang'a, Machakos, Nyandarua, and Nakuru Counties. 3700 HCWs of various cadres service the county public health system which is distributed among 449 public health facilities. These include three (3) Level 5 hospitals, forty-three (43) Level 4 facilities, twenty-eight (28) Level 3 facilities and three hundred and seventy-five (375) Level 2 facilities. Tigoni Level IV Hospital

was repurposed into the county's COVID-19 isolation centre. At the time of designing the study, none of the COVID-19 vaccines were available in Kenya.

## Study participants

In this study, a HCW included any person involved in the provision of health services to a user or those who are on facility grounds employed by the facility. At the time of the study, the county of Kiambu was served by a total of 3700 HCWs. These HCWs were divided into caregivers, administrative staff and environmental health workers, and details of each category can be found below.

1. Caregivers included all HCWs who interact with patients directly. They included: medical officers, consultants, nurses, clinical officers, dental officers, dental technologists, pharmacists, pharmaceutical staff, laboratory staff, orthopaedic technologist, nutritionists, radiographers, physiotherapists and mortuary attendants.

2. Administrative staff included HCWs who do not interact with patients directly. They included: health administrative officers and staff, health-supportive staff, medical engineering technologists, health records & information officers, medical social workers, ambulance drivers, and HIV testing services staff.

3. Environmental health workers consisted of public health staff, such as community health volunteers, health promotion officers, and public health officers/community health officers.

## Sample size and sampling

Cochrane's formula was used to determine the least possible sample size regarding the KAP of the HCWs based on a similar study published in Uganda where the proportion of sufficient KAP was 69%, 21% and 74%, respectively, as shown below:

$$n = \frac{Z^2 pq}{e^2}$$

We assumed a 95% confidence level (corresponding to a Z-score of 1.96) and a margin of error *e* of 5% (expressed as 0.05). Hence, at least 323 HCWs were needed for assessing knowledge, 257 to gauge attitudes, and 307 healthcare workers were required to evaluate practices. Therefore, we aimed for a sample size of at least 323 HCWs.

To ensure the sample's representativeness, specific cadres were weighted to align with the population. Caregivers were aimed to make up 78% of the sample; administrative staff were to comprise 12% of the sample, and environmental health workers were planned to account for 10% of the sample size. This sample was collected via volunteer purposive sampling, similar to another KAP study [19].

## Questionnaire

We employed a questionnaire from Olum *et al.* for this study [10]. This questionnaire was constructed as such:

1. Knowledge was assessed using a 11-item questionnaire adapted from Zhong *et al.* [20], and modified to suit HCWs, with each correct answer weighing one point. According to Zhong *et al.*, this was a reliable knowledge scale for adoption as it possessed a Cronbach's alpha score of 0.71 [20].
   We adopted this Knowledge scale for our study, however, we added three context-specific

questions regarding the location of Kiambu county's COVID-19 isolaton centre, handling a suspected COVID-19 case and data entry for suspected COVID-19 cases. These questions were selected and reviewed by public health experts.

2. Attitudes were assessed using 5 Likert-item questions that have been adopted from Goni *et al*. [21] and modified by Olum *et al* [10]. Goni *et al*.'s Cronbach's alpha scale measurement for this was 0.77 [21].
   We adopted this Attitude scale for our study, however we added six context-specific questions regarding how the national and county government structures were handling the pandemic at various stages: initially, currently and possibly in the future.

3. Practices were assessed using five Likert-item questions that have been developed from the WHO recommended practices. This was adopted fully from the Olum *et al* [10].
   The survey underwent a pre-test with research assistants and the study's principal investigator before the official data collection phase. After analyzing the pretest results, questions were reviewed and refined to enhance clarity, appropriateness, and eliminate redundancy. The alpha measurements for the scale subsets as used in our study were as follows: $\alpha_k$ = 0.69, $\alpha_a$ = 0.81 and $\alpha_p$ = 0.61. In this study, a Cronbach's alpha score of above 0.6 was considered adequate, similar to other KAP studies [19, 22–24]. The data underlying this research can be found in the S1 File.

## Variables of the study

The independent (predictor) variables of this study were socio-demographic factors (sex, age, facility, education status) and knowledge sources.

Knowledge scores were calculated by assigning 1 point to each correct answer, and a 0 to an incorrect/unknown answer. The total knowledge score ranged from 0 to 16, with higher scores signifying better knowledge. The total attitude score ranged from 0 to 15, and practices ranged from 0 to 6. There were three dependent variables in this study: the Knowledge, Attitudes and Practices (KAP) of HCWs. KAP scores were classified into three categories: poor, medium and good based on the Bloom's cutoff points: good (80–100%), moderate (60–79%) and poor (<60%) [25].

## Data collection

A semi-structured questionnaire was used for data collection and this paper will communicate the findings of the structured questionnaire alone. The findings of the qualitative data collected are discussed elsewhere [10]. The questionnaire was dispensed online from the 11th of March 2021, after which it was supplemented with hard copy questionnaires to meet the required sample sizes (1st June 2021 – 12th August 2021). Therefore, there was access to the questionnaire (either the soft or hard copy) from the 11th of March 2021 until the 12th of August 2021. None of the authors had access to the respondents' personal identifying information that could identify individual participants during or after data collection.

## Data analysis

Data cleaning and validation were done to ensure there were no duplicate responses.

Knowledge sources were classified as "Governing bodies of international health organizations (International Governing bodies)" which included the World Health Organization and Centre for Disease Control; "Social Media" for sites such as WhatsApp, Facebook, Twitter,

Instagram; and "Official government sites and media" included Ministry of health circulars. The categorical variables were presented as frequencies with percentages.

Multivariable Ordinal logistic regression models were constructed to determine the factors associated with good: (1) Knowledge, (2) Attitudes and (3) Practices of HCWs towards COVID-19. The models each assessed the associations of the independent variables with the KAP scores (dependent variables).

The link functions were chosen based on the changes in the cumulative probabilities: the "probit" link function for gradual changes (as noted with the knowledge and attitude types) and the complementary log-log link function for practice, for which the cumulative probabilities increased from 0 slowly and then rapidly approached 1 [26].

The crude odds ratio (COR) was obtained for each independent variable per dependent variable. These values can be found in the S1 Table. Independent variables with a COR p-value of $< 0.05$ was used to construct the initial fitted model. Then, a null model was run for each dependent variable. The results of ordinal logistic regression are not valid unless they satisfy Brant's test. To further validate the initial fitted models, each one underwent a check for satisfaction of the parallel regression/proportional odds assumption using the Brant test [27]. This posits that the slopes (represented by β-coefficients/odds ratios) of the model across various ordinal outcome categories remain constant, while the intercepts can vary [28]. The Brant test serves to evaluate the overall significance of the model as well as the individual significance of all explanatory variables included in the model, where a significant test result indicates the model is invalid [28].

The variables that did not satisfy Brant's test were removed and the final fitted model was obtained. The results were finally tabulated as the β-coefficient, adjusted odds ratios (AOR) at 95% confidence intervals (CI), and the p-value. Further details regarding each model can be found below.

Correlations between scores of KAP were also analyzed using Spearman's rank correlations given that the data were non-normally distributed. Correlations were interpreted using the following criteria:0–0.25 = weak correlation and 0.25–0.5 = fair correlation.

All analyses were done using R statistical software v.4.3.0 (R Foundation for Statistical Computing, Vienna, Austria). The level of significance was set at a *p*-value of $< 0.05$.

**KAP ordinal logistic regression modelling.** For knowledge, the null model was fitted (Aikake Information Criterion, AIC = 903.2; Residual Deviance: 899.1). The independent variables that had a significant COR were: sex, education, cadre, news, international, social media, medical fora and journals, and these were used to construct the initial fitted model. This initial fitted model was checked using Brant's test and predictor variables that failed the test were removed. As a result, cadre, news, international sites, social media and journals constituted the final model (AIC = 804.2; Residual Deviance: 788.2) which satisfied Brant's test ($X^2$ = 10.9, df = 6, p = 0.09).

For attitude, the null (AIC = 661.8; Residual Deviance: 657.8) was fitted and the model was saturated. The statistically significant explanatory variables were: education, cadre, government, social media and continuous medical fora as sources of information. The Brant's test indicated that education possessed a significant p-value. Thus, this predictor variable was dropped and final model fitted (AIC = 618.7; Residual Deviance: 604.7) which satisfied Brant's test ($X^2$ = 0.96, df = 5, p = 0.97).

For practice, the null model was run (AIC = 691.9; Residual Deviance: 687.9). The predictor variables with significant COR included: education, cadre, government, news and social media as information sources, and these were fitted into the initial fitted model. The Brant's test for the significant variables indicated that education was significant (p $< 0.05$). Thus, the final model was fitted (AIC = 631.7; Residual Deviance: 617.7), which satisfied Brant's test ($X^2$ = 4.63, df = 5, p = 0.46).

## Ethical considerations

Ethical approval was granted from the University of Eastern Africa, Baraton Ethics Review Committee (UEAB/REC/07/06/2020). No personal identifying information from the respondents was collected. All data was only available to the investigators. Participants were free to withdraw from the study at any time and participation was voluntary.

## Results

### Participant characteristics

Upon completion of the survey, 438 healthcare professionals responded: 71.0% (n = 315) used hardcopy questionnaires, while 29.0% (n = 129) used the online version.

Caregivers made up 73.5% (n = 322) of the total sample, administrative staff made up 15.1% (n = 66), and environmental health workers made up 11.4% (n = 50). For further information about the study participants categorized by cadre, please refer to the S2 Table. Majority of the caregivers (66.0%) had obtained a bachelor's degree, roughly half of the administrative staff (53.0%) had obtained a diploma, and around a third (32.0%) of the environmental health staff had obtained a certificate as the highest level of education. For information regarding COVID-19, over 70% of HCWs across all three categories accessed official government websites.

Based on the demographic features of the 438 respondents, 276 (64.5%) of those surveyed were female, and 144 (33.6%) were male (Table 1). The largest proportion of participants (20.5%) were in the 35–39 age range, while 20.2% were in the 30–34 age range. The greatest percentage of participants (28.1%) came from Level 4 health facilities, with Level 5 facilities coming in second with 27.6%. Most of the responders (59.6%) possessed a diploma, and 78.6% consulted official government sites for information regarding COVID-19. However, only 21.3% participated in continuous medical fora and 18.0% consulted medical journals.

### HCWs knowledge regarding COVID-19

The mean knowledge score was 11.524 (SD = 3.047, IQR = 9.00–14.00). 43.0% of the respondents had good knowledge, 23.0% had a medium grade of knowledge and 34.0% had poor knowledge (Fig 1A).

Regarding the main clinical symptoms of COVID-19, participants' responses are indicated in Fig 2. Fever (91.8%, n = 408), cough (84.2%, n = 374), and sore throat (65.9%, n = 293) were the most frequently selected symptoms.

The majority of the respondents concurred that there is no effective cure for COVID-19 and early management is crucial for patient recovery (80.5%), that respiratory droplets are the primary way that COVID-19 spreads (93.2%), wearing a general mask is preventive (85.3%) and isolation is effective in curbing the spread of the infection (92.5%).

The majority also rejected the following assertions: that the majority of patients (71.8%) will get a serious infection; that interaction with wild animals will cause infection (68.0%); that the virus cannot spread without a fever (70.7%); and that children do not need to follow preventive measures (79.7%) (Table 2).

Tigoni Level IV Hospital was accurately identified by 74.3% of participants (n = 330) as the county's COVID-19 isolation site. However, a sizable fraction selected the Thika level 5 hospital (20.5%, n = 91), and 4 individuals (0.9%) said they were unaware of the location of the county's COVID-19 isolation unit. Only 68.9% (n = 222) of the caregivers correctly recognised Tigoni Level IV Hospital when analysed specifically to cadre, compared to 84.8% (n = 56) of the administrative staff and 92.0% (n = 46) of the environmental staff.

**Table 1. Socio-demographic data of the respondents.** The n represents the number of valid responses for each particular variable.

| Variable | Label | Frequency | Percentage |
|---|---|---|---|
| Sex | | **n = 428** | |
| | Female | 276 | 64.5% |
| | Male | 144 | 33.6% |
| | Prefer not to say | 8 | 1.9% |
| Age | | **n = 441** | |
| | 20–24 | 5 | 1.1% |
| | 25–29 | 71 | 16.1% |
| | 30–34 | 89 | 20.2% |
| | 35–39 | **91** | **20.5%** |
| | 40–44 | 79 | 17.9% |
| | 45–49 | 47 | 10.7% |
| | 50–54 | 27 | 6.1% |
| | 55–59 | 32 | 7.3% |
| Facility | | **n = 424** | |
| | Level 1 | 9 | 2.1% |
| | Level 2 | 49 | 11.6% |
| | Level 3 | 84 | 19.8% |
| | Level 4 | **119** | **28.1%** |
| | Level 5 | 117 | 27.6% |
| | Sub-county official | 29 | 6.8% |
| | County official | 17 | 4.0% |
| Education | | **n = 441** | |
| | Certificate | 43 | 9.8% |
| | Diploma | **263** | **59.6%** |
| | Bachelors | 84 | 19.0% |
| | Masters | 40 | 9.1% |
| | PhD | 6 | 1.4% |
| | Other | 5 | 1.1% |
| Knowledge sources | | | |
| | Official government sites | **349** | **78.6%** |
| | News media | 254 | 57.6% |
| | International Governing bodies | 223 | 50.2% |
| | Social media sites | 192 | 43.2% |
| | Continuous medical fora | 95 | 21.3% |
| | Medical journals | 80 | 18.0% |

HCWs who correctly identified the appropriate actions when encountering a patient with a high suspicion of Covid-19 at work are detailed in Table 3.

Caregivers specifically were probed regarding the entry of data for a suspected COVID-19 case (correct responses were case investigation form and patient notes). For this question, 167 respondents (51.8%) indicated using the case investigation form, 102 (31.7%) preferred the clinical register, 63 (19.6%) opted for the case definition form, and 62 (19.2%) selected patient notes. Additionally, 10 respondents (3.1%) admitted not knowing the correct method. Among these responses, 229 (71.1%) chose at least one correct option, while 25 (7.9%) accurately selected both correct answers (Table 3).

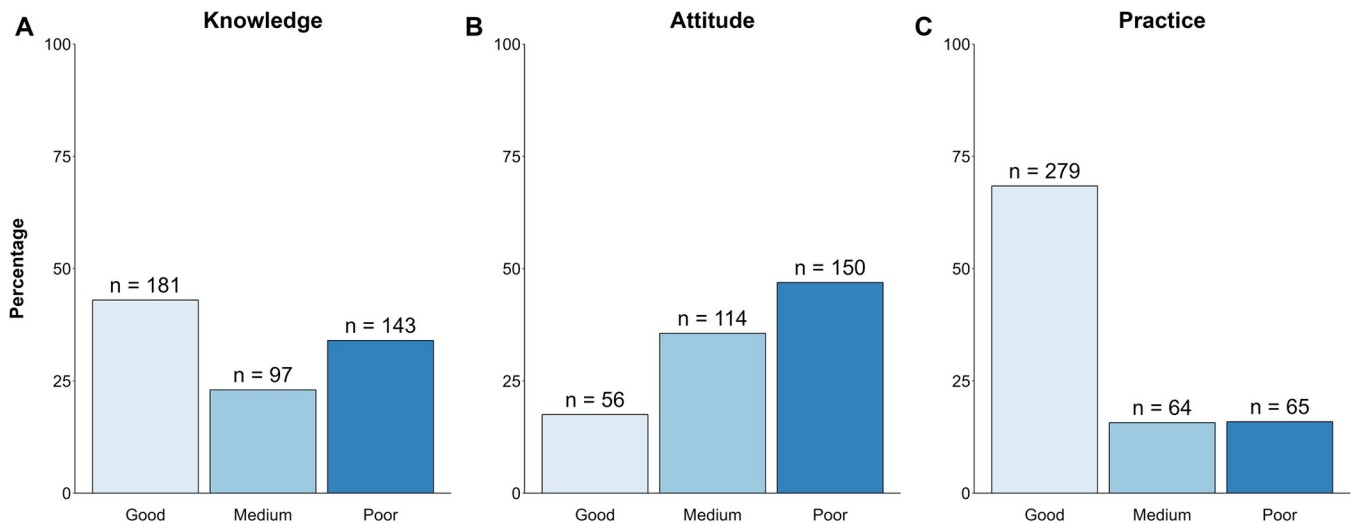

**Fig 1. The distribution of good, medium, and poor knowledge, attitudes, and practices as determined in this study, categorized based on Bloom's cutoff points.**

## The attitudes of HCWs towards COVID-19

The mean attitude score was 8.596 (SD = 3.714, IQR = 6.000–12.000). Only 17.5% of the respondents had a good attitude, 35.6% had a medium grade attitude and 46.9% had a poor attitude (Fig 1B).

Almost half of the respondents (49.0%) strongly disagreed that black race confers protection from COVID-19. The attitudes towards the effectiveness of preventative practices such as wearing a mask (36.7%) and using soap for handwashing (39.7%) were mostly affirmative

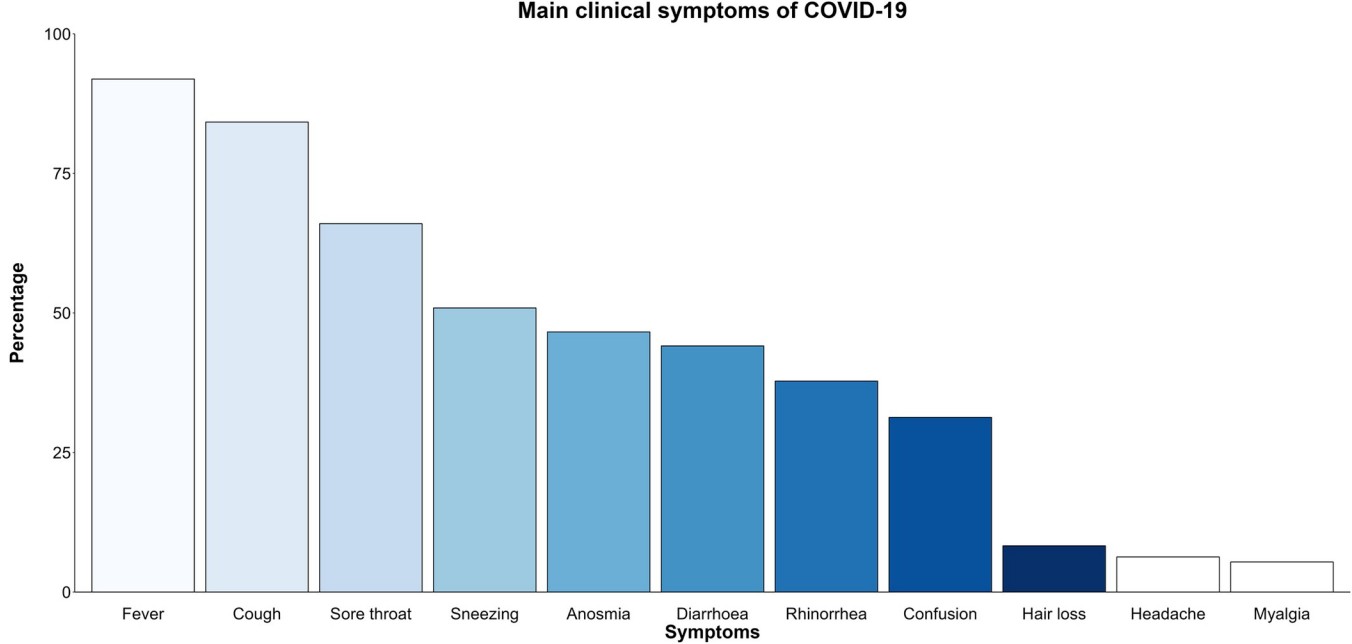

**Fig 2. Responses to the main clinical symptoms of COVID-19 from all respondents.**

**Table 2. Frequency of the responses to the knowledge questions asked to all respondents.** Expected answers are provided at the end of the question in italics. The numbers within the brackets indicate the frequency of the response.

| Knowledge Questions | True | False | I don't know |
|---|---|---|---|
| There is currently no effective cure for COVID-19, but early treatment can help most patients recover. (T) | **355 (80.5%)** | 82 (18.6%) | 4 (0.9%) |
| Most persons with COVID-19 will develop severe cases (F) | 118 (26.6%) | **318 (71.8%)** | 7 (1.6%) |
| Eating/being in contact with wild animals would result in the COVID-19 virus infection. (F) | 117 (26.4%) | **302 (68.0%)** | 25 (5.6%) |
| Persons with COVID-19 cannot transmit the virus to others when a fever is not present. (F) | 119 (26.9%) | **313 (70.7%)** | 11 (2.5%) |
| The COVID-19 virus spreads via respiratory droplets of infected individuals. (T) | **413 (93.2%)** | 24 (5.4%) | 6 (1.4%) |
| Wearing general medical masks is preventative against COVID-19 virus. (T) | **377 (85.3%)** | 59 (13.3%) | 6 (1.4%) |
| It is unnecessary for children and young adults to take measures to prevent infection. (F) | 87 (19.6%) | **354 (79.7%)** | 3 (0.7%) |
| Isolation and treatment of people who are infected with COVID-19 virus are effective. (T) | **407 (92.5%)** | 30 (6.8%) | 3 (0.7%) |

(Table 4). When it came to treating patients who had symptoms and signs that might point to COVID-19, 40.7% of respondents said they felt confident. However, the general attitude towards their own personal safety, their training regarding personal protective equipment, and the process of producing vaccines (17.8%, 23.8%, and 14.8% strongly agreed) were less compelling. Over 33% of respondents said that both the initial and ongoing management of COVID-19 was sufficient, and that Kenya and Kiambu County are in a strong position to contain the virus. Only 4.4% of respondents strongly agreed, meanwhile, that they thought the community members were well-informed about COVID-19.

When asked if they would be willing to have the vaccine whenever it became available, 69.5% of HCWs gave an affirmative response, 21.1% gave a hesitant reaction (a "Maybe"), and 9.3% gave a negative response (a "No").

## HCWs COVID-19 practices

The mean practice score of caregivers on COVID-19 was 4.588 (SD = 0.899, IQR = 4.000–5.000). Of the responses, 68.4% of the HCWs had good practices, 15.7% had a medium practice and 15.9% had poor practices (Fig 1C).

When interacting with a patient, nearly all of the caregivers disclosed always using a mask (97.8%) and never shaking hands (96.6%) (Table 5). While most caregivers (89.8%) always cleaned their hands after treating a patient, only 77.6% of them took the time to educate patients about COVID-19. The majority of caregivers always avoided patients suspected of being infected with COVID-19 (69.3%).

**Table 3. Table depicting the responses of the HCWs regarding the set protocol observed when they encounter a patient with a high index of suspicion for COVID-19 while at work.** There were 3 correct answers in the choices. The raw number of responses are shown within the brackets.

| | Caregivers | Administrative staff | Environmental health staff |
|---|---|---|---|
| **Incorrect responses** | 118 (36.6%) | 16 (24.2%) | 13 (26.0%) |
| **1 correct response** | 93 (28.9%) | 18 (27.3%) | 9 (18.0%) |
| **2 correct responses** | 47 (14.6%) | 14 (21.2%) | 13 (26.0%) |
| **All 3 correct responses were selected** | 64 (19.9%) | 18 (27.3%) | 15 (30.0%) |

**Table 4. Attitude questions and their respective responses from all the health care workers.** The numbers within the brackets indicate the frequency of the response.

| | Strongly agree | Agree | Not sure | Disagree | Strongly disagree |
|---|---|---|---|---|---|
| Black race is protective towards COVID-19 disease. | 17 (3.9%) | 33 (7.5%) | 66 (15.0%) | 102 (23.1%) | **216 (49.0%)** |
| Wearing a well-fitted masks is effective in preventing COVID-19. | 155 (35.1%) | **162 (36.7%)** | 43 (9.7%) | 15 (3.4%) | 67 (15.2%) |
| Using normal soap for hand washing can prevent you from getting COVID-19 | 140 (31.75%) | **175 (39.7%)** | 55 (12.5%) | 16 (3.6% | 55 (12.5%) |
| When a patient has signs and symptoms of COVID-19, I can confidently participate in the management of the patient. | 93 (21.7%) | **174 (40.7%** | 68 (15.9%) | 45 (10.5%) | 48 (11.2%) |
| In my everyday work, I feel confident that I am safe and protected from contracting COVID-19. | 78 (17.8%) | **111 (25.3%)** | 76 (17.3%) | 93 (21.2%) | 81 (18.5%) |
| I am well trained on the use of PPE to protect me from contracting COVID-19. | 103 (23.8%) | **166 (38.4%)** | 26 (6.0%) | 72 (16.7%) | 65 (15.0%) |
| I am confident about the vaccine production process for the COVID-19 vaccine. | 62 (14.8%) | **128 (30.5%)** | 93 (22.2%) | 41 (9.8%) | 95 (22.7%) |
| From my interaction with patients, I believe the Kiambu county community is well sensitized on COVID-19. | 19 (4.4%) | **156 (36.0%)** | 76 (17.6%) | 58 (13.4%) | 124 (28.6%) |

## Factors associated with KAP

Table 6 presents the results of three multivariable ordinal logistic regression final fitted models on the factors associated with the knowledge, attitudes and practices of HCWs towards COVID-19. Compared to being an administrative staff member, being a caregiver was associated with a reduced odds of having a higher knowledge score (p = 0.0002). However, there was a significant positive association between obtaining information from international health sites (p = 0.0178), social media sites (p = 0.0034), and journals (p = 0.0002) with obtaining a higher knowledge score.

Compared to being an administrative staff member, being a caregiver or an environmental health staff member was associated with an increased odds of having a better attitude (p = 0.0253 and 0.0062, respectively). While consulting continuous medica fora decreased the odds of having a good attitude significantly (p = 0.0156), referring to government sites for information regarding COVID-19 produced better attitudes, and this was statistically significant (p = 0.0036).

**Table 5. Caregivers' responses on the various practice questions.** Environmental health staff and administrative staff were excluded for this analysis. The numbers within the brackets indicate the frequency of the response.

| | Always | Never | Occasional |
|---|---|---|---|
| In the last 1 week, I have worn a mask when in contact with patients. | **309 (97.8%)** | 1 (0.3%) | 6 (1.9%) |
| In the last 1 week, I have worn PPE when in contact with patients. | **241 (76.8%)** | 35 (11.1%) | 38 (12.1%) |
| In the last 1 week, I have refrained from shaking hands. | **309 (96.6%)** | 3 (0.9%) | 8 (2.5%) |
| In the last 1 week, I have washed my hands before and after handling each patient. | **283 (89.8%)** | 4 (1.3%) | 28 (8.9%) |
| In the last 1 week, I have avoided patients with signs and symptoms suggestive of COVID-19. | **217 (69.3%)** | 60 (19.2%) | 36 (11.5%) |
| In the last 1 week, I have taken time to sensitize patients and their families on COVID-19 | **246 (77.6%)** | 15 (4.7%) | 56 (17.7%) |

**Table 6. Results of the ordinal logistic regression analysis of factors associated with knowledge, attitudes and practices of HCWs in Kiambu county, Kenya towards COVID-19.**

| Dependent variable | Independent variable | Label | Coefficient (β) | Adjusted odds ratio | p-value |
|---|---|---|---|---|---|
| **Knowledge** | | | | | |
| | Cadre | | | | |
| | | Administrative staff | | Ref | |
| | | Caregivers | -0.644 | 0.525 (0.371–0.738) | **0.0002** |
| | | Environmental health staff | -0.076 | 0.926 (0.586–1.465) | 0.7440 |
| | Sources of information | | | | |
| | | News media | -0.102 | 0.902 (0.677–1.202) | 0.4855 |
| | | International Governing bodies | 0.324 | 1.382 (1.058–1.807) | **0.0178** |
| | | Social media | 0.409 | 1.505 (1.145–1.979) | **0.0034** |
| | | Medical journals | 0.642 | 1.899 (1.352–2.683) | **0.0002** |
| **Attitudes** | | | | | |
| | Cadre | | | | |
| | | Administrative staff | | Ref | |
| | | Caregivers | 0.454 | 1.574 (1.062–2.355) | **0.0253** |
| | | Environmental health staff | 0.709 | 2.032 (1.225–3.385) | **0.0062** |
| | Sources of information | | | | |
| | | Social media | -0.241 | 0.785 (0.603–1.023) | 0.0735 |
| | | Continuous medical fora | -0.383 | 0.681 (0.499–0.928) | **0.0156** |
| | | Official government sites | 0.536 | 1.710 (1.194–2.465) | **0.0036** |
| **Practice** | | | | | |
| | Cadre | | | | |
| | | Administrative staff | | Ref | |
| | | Caregivers | 0.754 | 2.125 (1.322–3.299) | **0.0012** |
| | | Environmental health staff | 0.303 | 1.353 (0.738–2.534) | 0.3322 |
| | Sources of information | | | | |
| | | Official government sites | 0.588 | 1.800 (1.181–2.681) | **0.0048** |
| | | News media | -0.153 | 0.858 (0.555–1.312) | 0.4834 |
| | | Social media | -0.930 | 0.395 (0.261–0.590) | **$7.8 \times 10^{-6}$** |

Compared to administrators, caregivers had approximately 2-fold greater odds of having better practice (p = 0.0012). Consulting government sites for information about COVID-19 had almost twice the odds of having better practice (p = 0.0048), while consulting social media sites reduced the odds of having good practice (p = 0.0000078).

## Correlation between knowledge, attitude and practice scores

There was a statistically significant fair correlation between knowledge and practice ($r$ = -0.3300, p = $2.766 \times 10^{-11}$), a weak correlation between knowledge and attitude ($r$ = -0.1541, p = $6.538 \times 10^{-3}$) and a fair correlation between attitude and practice toward COVID-19 disease ($r$ = 0.3443, p = $6.843 \times 10^{-10}$). These correlations can be found in Fig 3.

## Discussion

This study was conducted during the third wave of the COVID-19 outbreak in Kenya observed from March to April 2021 [18]. It is crucial and imperative to obtain the perspectives of those who disproportionately bore the brunt of the pandemic. This analysis of HCWs' knowledge

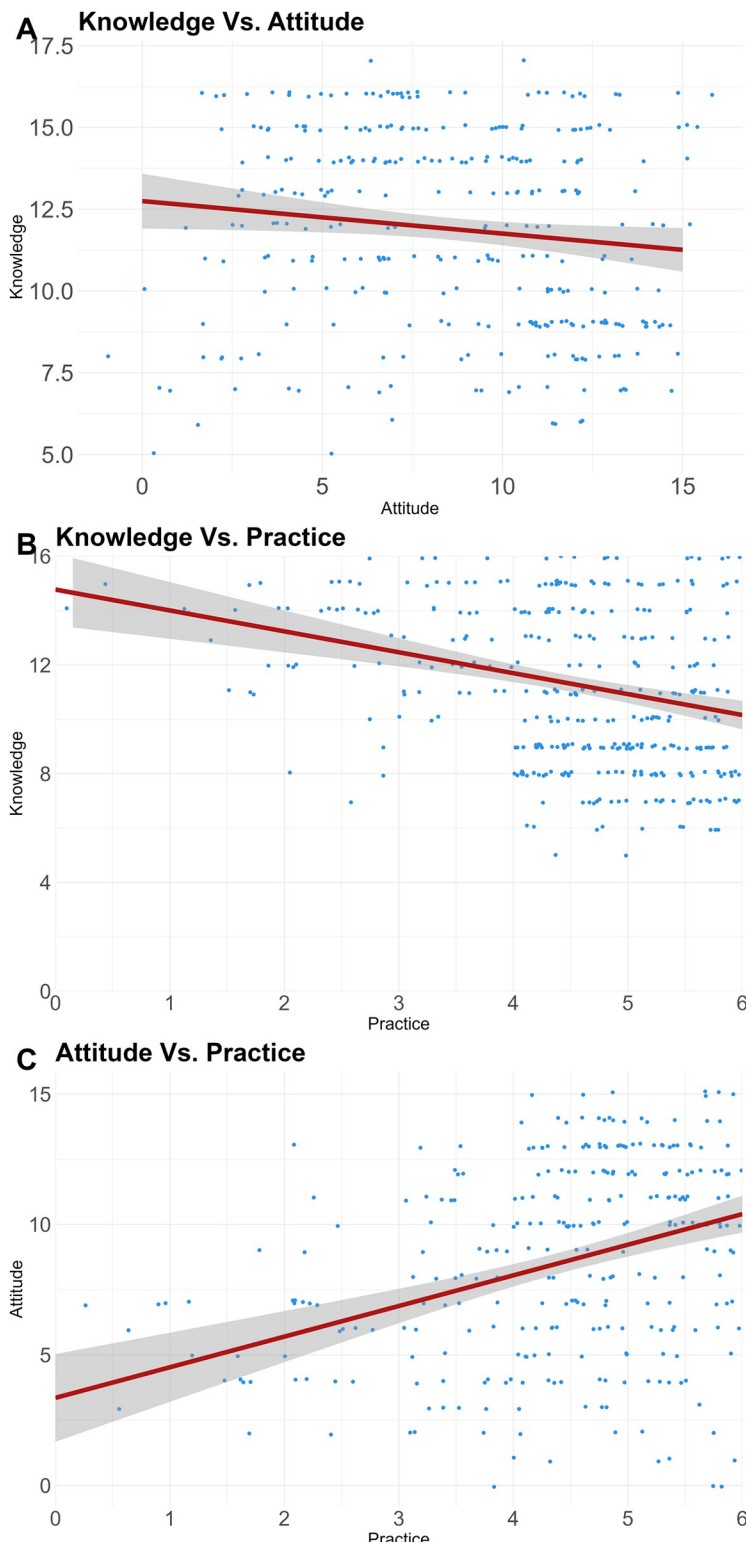

**Fig 3.** A–a line graph depicting the relationship between knowledge and attitude, B–knowledge and practice and C–attitude and practice with a line-of-best fit.

and the factors affecting their attitudes and practices could provide a contextual reference for future public health crises of similar proportions that affect HCWs.

The majority of the respondents in the current study were female and between 35–39 years old with majority holding a diploma. This is in contrast to a similar study conducted in Makerere University Teaching Hospitals in Uganda whereby 64% of the participants were male and the median age was 34 years old with 92% having a bachelor's degree [10]. The majority of the respondents were caregivers. This is in contrast to a study conducted in Henan, China whereby 57.4% of the HCWs surveyed were non-frontline [29]. Government sites were used as the source of information for COVID-19 by 78.6% of the respondents. This is in contrast to a study conducted in Uganda where the most common source of information was an international health organization [10].

## Knowledge

Less than half (43.0%) of the surveyed respondents in this study had good knowledge regarding COVID-19. This is in contrast to a study conducted in Ethiopia whereby 88.2% had good knowledge about COVID-19 and a study in China where 90% of HCWs had good knowledge [20, 30]. Our finding may be due to the ever-evolving nature of accurate knowledge during the pandemic as a result of the rapid paradigm shifts in public health advisories. However, this still remains a staggering result, given that imperative knowledge gaps regarding patient management protocols also emerged, particularly in caregivers tasked with the direct management of patients where only 19.9% of the caregivers indicated the correct operating procedures set out at the time.

Another worrying result was the fact that almost a quarter of the respondents did not correctly identify the COVID-19 isolation. Both of these variable proportions were abysmal in the caregivers specifically. Furthermore, only 7.9% of the caregivers knew the specific documents to input data regarding suspected COVID-19 patients. These are critical information gaps that should not be so wide, especially for frontliners in the midst of a health crisis.

The respondents in the current study indicated fever, cough, and sore throat as the main clinical symptoms of COVID-19. This is unlike a KAP study conducted in Sierra Leone among HCWs who indicated the main symptoms as fever, fatigue, dry cough, and myalgia [6]. This discrepancy may be due to the different time periods the questionnaires were administered as the main clinical symptoms of COVID-19 did evolve as the pandemic progressed and new variants emerged [31, 32]. Given that our study was conducted during the third wave of COVID-19 in Kenya [18], the Alpha variant B.1.1.7 was rampant at this time (since January 2021) with the main symptoms being: cough, ageusia, fever and appetite loss [18, 31].

In this study, approximately 80% of the HCWs gave the correct answers to the questions regarding prevention and general knowledge about COVID-19. These responses mirror those in a study conducted in Uganda from which the tool was adopted from [10] and highlight the similar level of training between HCWs on COVID-19. However, 85.3% indicated that wearing general masks prevents infection and isolation is an effective way to treat infected people, but this assertion and the type of mask that is protective has been subject to debate [10, 33, 34].

The odds of having good knowledge, overall, were higher in caregiving staff. However, the specific questions regarding patient management protocols (besides clinical management) were performed abysmally in the caregivers. This is in contrast with a study conducted in Uganda among HCWs where their specific cadre did not determine their knowledge of COVID-19 [10]. Obtaining information about COVID-19 from international government sites, social media and journals were all associated with better knowledge scores. This is similar to a study conducted by Jemal et al. in HCWs in a multicenter study in Ethiopia [35]. These

findings highlight the necessity of a quick (social media) and convenient (social media and international government sites) dissemination of accurate information. This approach could be crucial for sharing updates on developments, new precautions, additional research findings, and updated protocols. In the future, combining these sources of information can be employed to provide HCWs with specific information about emerging health challenges.

## Attitude

Only 17.5% of HCWs had a good attitude in this study. Most of the HCWs in Kiambu county had a poor attitude in regards to COVID-19. This is congruent with studies conducted in Ethiopia during the Ebola pandemic [36] and a KAP study conducted at Makerere University Teaching Hospital [10]. A considerable proportion of HCWs were not confident in managing COVID-19 patients, did not feel protected from contracting COVID-19 while working and did not feel well trained in the use of PPEs. This is in contrast to a study conducted in Bangladesh where 88.8% of HCWs had a positive attitude regarding PPE use [37]. HCWs can only effectively prevent infection by mastering and consistently demonstrating the correct procedures for donning, doffing, and using PPE [37]. Therefore, this result highlights a gap in efficacy beliefs regarding PPEs in the HCWs of Kiambu county, indicating a potential area for improvement in their understanding and implementation of preventive measures. However, the attitude toward accepting the vaccine once made available was positive and this is in line with a study conducted in Western Ethiopia where the majority of HCWs held favorable views on COVID-19 vaccination [38].

Social media sites and medical fora were associated with increased odds of having a poor attitude. This is in line with studies correlating social media use to worsening attitude and mental health, even without the context of a pandemic [13, 39]. Social media users are more likely to heavily consume news stories [39] perhaps due to the explosive dissemination of posts [13]. In addition, the chorus of contrasting voices on social media contributing to information and opinion overload makes strengthening one's resolve and attitude becomes all the more difficult [13, 40]. These factors could have produced the association found in this study.

The factors associated with increased odds of a good attitude included obtaining information about COVID-19 from government sites. This is in contrast to the Ugandan study where HCWs who obtained information from mainstream media were four-times more likely to have a good attitude [10].

## Practice

Our study shows that most of the HCWs in Kiambu County had good preventative practices against COVID-19. Similar findings have been reported in HCWs on COVID-19 in both Sierra Leone [6] and Saudi Arabia [41]. Majority of the HCWs were adherent to the prevention measures recommended by the Ministry of Health Kenya and the World Health Organization. However, 66.3% avoided patients with signs and symptoms of COVID-19, which further indicates that, regardless of their compliance to the preventive protocols, the HCWs were not confident in their efficacy.

Caregivers (compared to administrative staff) were two times more likely to have correct practices. This is likely due to their direct role in patient care compared to administrative staff, making the correct practices more crucial. Receiving information from the government increased the odds of good practice. Furthermore, social media reduced the odds of having good practice. A study conducted in Uganda produced different findings [10]. This incongruence may be a result of social media and information overload causing increased consumption of disinformation and conflicting advisory practices from the internet.

## KAP relationship

In our study, we observed an inverse relationship between the scores for knowledge, attitudes, and practices among HCWs. Surprisingly, higher knowledge scores were associated with less favourable attitudes and practices among HCWs. This inverse relationship was notably stronger between knowledge and practice. However, we found that a positive attitude was directly correlated with the adoption of correct practices. These observations align with findings reported in a study conducted in Ethiopia [7]. However, these relationships, particularly between knowledge and attitude and knowledge and practice, are in contrast with studies from Nepal and Uttar Pradesh [8, 42]. This is a peculiar finding as convention dictates that the more one knows the better their practices and attitude. Therefore, in our context, regardless of acquiring knowledge about COVID-19, this did not further translate into optimal practice or a positive attitude. This contradicts evidence that efficacy beliefs predict behaviour, which has been shown for a myriad of different diseases [43–47]. Therefore, in addition to being knowledgeable, Kiambu County HCWs needed to be further convinced that the practices for its prevention would be effective.

## Recommendations

International governing bodies and local government sites emerged as sources of information that produce better knowledge; and better attitudes and better practices, respectively. This study calls for the improvement of communication strategies within the healthcare system. We advocate for the strengthening of the communication channels between International governing bodies, local government and the HCWs, especially in times of crises. These channels should be moderated and frequently updated by public health officials to ensure the achievement of the best possible KAP in this vulnerable population and to address the inverse KAP relationship observed in this study. This could be supported by a comprehensive online public health database that provides up-to-date, essential information regarding emerging diseases, such as the symptomatology, management, isolation protocols and data entry. We recommend upscaling the collaborative efforts between international health governing bodies and local governments in providing pertinent data regarding a health problem. The local governments should then constantly and consistently update HCWs regarding new evidence and updated patient management protocols.

While social media produced good knowledge, it did so at the detriment of the HCW's attitude and practices. Given the widespread use and popularity of social media, addressing this could pose a challenge. The bad attitude could be as a result of viral dissemination of potentially upsetting material via social media (especially during crises) and due to the nature of HCWs' work which already poses a mental health challenge [48]. However, advisories in health facilities encouraging HCWs to take regular social media breaks could be helpful, as could limiting facility Internet access to just the pertinent Government websites. Furthermore, HCWs could be advised to seek out information from Government sources (which produced better attitudes and practices), and perhaps these can be disseminated via alternate means (via text messages, emails or facility pamphlets).

Identifying a substantial number of HCWs lacking confidence in managing COVID-19 patients and feeling unprotected highlights a clear area for improvement in their understanding and implementation of preventive measures. Interestingly, the study shows that government information sites played a crucial role in shaping positive attitudes among HCWs and it was the most common source of information regarding COVID-19. This underscores the weight of the government's voice in times of crises.

In addition to this, HCWs must be well-equipped. The fear involved in managing COVID-19 cases and avoiding patients could have also been attributed to the PPE shortage [16], which

were vital to protect the HCW and instil within them the confidence to carry out their duties. This speaks more to pandemic preparedness and the creation of a repository of medical equipment that, ideally, should be locally produced to meet urgent needs [49].

Finally, policymakers and stakeholders can establish an in-house system for continuous support and feedback from HCWs. HCWs should have a platform to voice their concerns directly, ask questions, raise concerns and receive guidance where doubt exists regarding protocols. This can also potentially remedy the inverse KAP relationships shown in this study. Regular feedback sessions within health facilities can allow for timely adjustments and improvements in training programs for future health crises.

## Limitations

This study has limitations, such as being cross-sectional, which prevented a more in-depth exploration of each KAP item. We also did not take into account the experience of HCWs in the private sector. The sampling strategy was also voluntary purposive sampling. Additional characteristics linked with COVID-19 behaviours such as perceived obstacles or other communication aspects were not evaluated in this study. Similar to other surveys, ours may also be affected by response bias. However, this study provided vital information on the KAP of HCWs in the Kenyan context.

## Conclusion

This study has important contextual implications. This undermines the notion that beliefs about effectiveness alone may accurately predict preventative behaviors. The study emphasizes the substantial influence of government websites (both at the international and local level) in generating favorable KAP in the context of a pandemic in Kiambu county, Kenya. This research advocates for better communication strategies within the healthcare system and suggests establishing public health platforms tailored to HCWs to deliver this information expeditiously. Furthermore, combining government information sites with medical forums and social media breaks can potentially address the inverse KAP relationship in our study. This can also be tackled at the regional healthcare system level through the implementation of a structured framework for ongoing assistance, constructive feedback, and educational initiatives aimed at enhancing future crisis management. By taking these measures, better support and protection can be afforded to HCWs, ultimately bolstering our preparedness and response to address forthcoming health crises.

## Supporting information

**S1 File. Collected data for this study.** This includes all the quantitative data collected for this study.
(XLSX)

**S1 Table. Crude odds ratio and the p-values.** This is a summary table of all the KAP crude odds ratio and their respective p-values used to construct the initial fitted model for ordinal logistic regression.
(PDF)

**S2 Table. Table of breakdown of socio-demographic characteristics of caregivers, administrative staff and environmental health staff.** Additional information regarding the study participants.
(PDF)

## Acknowledgments

We would like to acknowledge Stephen Musyoka, David Ndegwa, Maxwell Murage, Benson Mwaniki, Janefer Nyawira, Patrick Nyaga and Joseph Murega from the Kiambu County Government, Kenya, for their technical and logistical support to conduct this study.

## Author Contributions

**Conceptualization:** Prabhjot Kaur Juttla, Moses Ndiritu, Magoma Mwancha-Kwasa.

**Data curation:** Prabhjot Kaur Juttla, Moses Ndiritu, Ferdinand Milliano, Alfred Owino Odongo, Magoma Mwancha-Kwasa.

**Formal analysis:** Prabhjot Kaur Juttla, Moses Ndiritu, Magoma Mwancha-Kwasa.

**Investigation:** Moses Ndiritu, Ferdinand Milliano, Alfred Owino Odongo, Magoma Mwancha-Kwasa.

**Methodology:** Moses Ndiritu, Ferdinand Milliano, Alfred Owino Odongo, Magoma Mwancha-Kwasa.

**Project administration:** Moses Ndiritu, Ferdinand Milliano, Magoma Mwancha-Kwasa.

**Resources:** Moses Ndiritu, Ferdinand Milliano, Magoma Mwancha-Kwasa.

**Supervision:** Moses Ndiritu, Alfred Owino Odongo, Magoma Mwancha-Kwasa.

**Validation:** Prabhjot Kaur Juttla, Ferdinand Milliano, Alfred Owino Odongo, Magoma Mwancha-Kwasa.

**Visualization:** Prabhjot Kaur Juttla.

**Writing – original draft:** Prabhjot Kaur Juttla, Alfred Owino Odongo, Magoma Mwancha-Kwasa.

**Writing – review & editing:** Prabhjot Kaur Juttla, Moses Ndiritu, Ferdinand Milliano, Alfred Owino Odongo, Magoma Mwancha-Kwasa.

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
