## [Decision Letter · Decision Letter 0]

13 Oct 2023

PONE-D-23-19304Practice What You *Treat*: Knowledge, Attitude, and Practices of Healthcare Workers during a Pandemic in a Low-Resource SettingPLOS ONE

Dear Dr. Juttla,

Thank you for submitting your manuscript to PLOS ONE. After careful consideration, we feel that it has merit but does not fully meet PLOS ONE’s publication criteria as it currently stands. Therefore, we invite you to submit a revised version of the manuscript that addresses the points raised during the review process.

We look forward to receiving your revised manuscript.

Kind regards,

Jianhong Zhou

Staff Editor

PLOS ONE

Journal Requirements:

Reviewers' comments:

Reviewer's Responses to Questions

**Comments to the Author**

1. Is the manuscript technically sound, and do the data support the conclusions?

Reviewer #1: Yes

Reviewer #2: Partly

2. Has the statistical analysis been performed appropriately and rigorously? 

Reviewer #1: Yes

Reviewer #2: No

3. Have the authors made all data underlying the findings in their manuscript fully available?

Reviewer #1: Yes

Reviewer #2: No

4. Is the manuscript presented in an intelligible fashion and written in standard English?

Reviewer #1: Yes

Reviewer #2: No

5. Review Comments to the Author

Reviewer #1: The authors' primary focus was on exploring the knowledge, attitudes, and practices of healthcare workers during the Covid-19 pandemic. The paper is engaging and generally well-written. After thoroughly reading the manuscript, I have the following feedback:

Abstract :

• Their objective does not center on exploring the KAP. The term "evaluating" suggests an assessment of interventions, treatments, or comparisons between pre and post studies.

• They utilized multivariate logistic regression to adjust for other variables and estimate the adjusted odds ratio (AOR) associated with a 95% confidence interval. It might be better to incorporate these details in the methods section.

• Instead of presenting p-values as "p-value <0.001," it would be clearer to state them as "p=xxxe-10."

• Could you provide information on the number of healthcare workers with good/bad practices?

• In conclusion, the authors did not mention the associated factors that need improvement for raising awareness.

Methods:

• The authors briefly mention the study design, but a more detailed explanation is necessary.

• The sample size calculation provided is quite general and lacks important information, such as the proportion (p) of healthcare workers in the study. A 50% proportion is not appropriate for this type of study. Based on previous research, the authors should calculate the sample size specifically for healthcare workers (HCWs).

• It is stated that this is an online survey, and the authors mention that the survey was conducted until they reached a sample size of 385 HCWs. However, it appears that this survey did not take the sample proportionally, and further clarification is needed.

Results:

• In lines 146-147, it would be helpful to provide the number of indicators and the total score of knowledge for better clarification.

• In line 189, it is not possible for the 95% confidence interval of the odds ratio (OR) to be negative (-0.05 to 0.15).

• The study suggests an inverse association between practices and attitudes with knowledge. Is this interpretation accurate?

Discussion:

• In lines 241-244, the authors suggest that there were more gaps in knowledge and practices among HCWs. They found that HCWs did not exhibit appropriate practices, yet they had sufficient knowledge. However, their data analysis did not support this observation, as it showed that HCWs with insufficient knowledge had good practices. These results appear contradictory.

• In line 277, a major limitation of the study was its cross-sectional, which prevented the authors from conducting a more in-depth exploration of the KAP items.

• While the authors presented a well-written conclusion, it would be beneficial to provide further clarification on the factors that need improvement and how to address them.

Reviewer #2: Title

The title is not easy to understand for the readers

Abstract

• Background is well written

• The study period (May to August) is not consistent with the mentioned period in the study design on page 4. The questionnaire was used to collect data not for determining associated factors of KAP. Simple and multiple logistic regression, not multivariate logistic regression. See also comments in the methods and materials section

• Results should be revised according to the comments raised in the methods and materials sections

• ‘Messenger’ word should be rephrased; authors used this term in the whole manuscript

Keyword

• Most of the keywords are overlapped with the title of this study

Introduction

• The introduction section is not shaped and written as required for scientific research

• The authors did not consider quality points to highlight why they conducted this study, and about research gaps

• Need a comprehensive review of relevant literature

• The objective of this study is not clearly mentioned

• What are the implications of the findings of this study

Methods and Materials

• The definition of the study population is not clear

• It is also not clear, who are actually HCWs in this study or the way authors defined HCW, required quality modification

• According to the nature of the study setting and study population, the sample size determination and sampling technique are inappropriate. The sampling on a volunteer basis is not scientifically robust

• It is confusing, what type of questionnaire the authors used to collect the data: structured/open-ended/semi-structured? How could a questionnaire be structured and open-ended simultaneously? Probably, the authors considered a semi-structured research tool

• The authors did not mention how they checked the reliability, validity, and consistency of their data and questionnaire

• The authors did not mention what was the basis of the classification of the target variables

• The authors did not check the bivariate association of the target variables with the predictor variables, so, how did they select probable predictor variables for multiple regression modeling? In the modeling, some potential predictors may be insignificant due to arbitrary selection of the predictors

• The authors actually performed binary and multiple logistic regression but they mentioned “multivariate logistic regression”

• How the authors selected the link function for logistic regression as they did not check the nature of the target variable

Results

• The results should be revised according to the comments in the methodology section

Authors should also consider the following specific observations:

• The results are poorly written and presented

• Tables are too long

• Authors unnecessarily mentioned too long questions in the text

• Authors used ‘n’ for indicating sample size in the methodology section; in the result section it is confusing about the notation ‘n’ and ‘N’

• In Tables 4, 6, and 8, the authors provided a column ‘p-value’. P-value of which estimates: crude odds ratio (COR) or adjusted odds ratio (AOR)? If it indicates AOR, then where is the significance value of the COR estimates? Without significance value, how these results could be interpretable?

• It is clear from Tables 4, 6, and 8, that the authors performed simple logistic regression for all predictor variables as well as performed multiple logistic regression including all the predictor variables in the model, Why? Why did the authors consider the predictors with insignificant COR in the multiple logistic regression modeling?

• Estimates and p-values are not presented in a unique and standard format

• The second column of Tables 4, 6, and 8 should be renamed as ‘Label’

• In Table 4, the authors provided COR for attitude and practice considering these as predictor variables of knowledge. In principle, how attitude and practice could be the predictors of knowledge? Is there any basis or reference against this issue? If yes, why there is no adjusted odds ratio for these factors? Probably, the authors considered it wrongly. Authors should rethink and redesign

• In Tables 6 and 8, the authors did not consider the knowledge level as a predictor variable for attitude and practice, why?

• Is the result of correlation measurement between knowledge and practice correct? If yes, does this type of relationship exist in reality? The authors did not discuss the reason for this result in the discussion section. What would be the implications of this finding?

Discussion

• The discussion section is not written as expected for a scientific article

• The authors are suggested to revise the methodology and results according to the comments raised in the respective sections

• They should rewrite this section by explaining the potential findings with proper supporting citations

• Authors are also suggested to add implications of the findings and limitation of the study

6. PLOS authors have the option to publish the peer review history of their article (what does this mean?). If published, this will include your full peer review and any attached files.

Reviewer #1: No

Reviewer #2: No

---

## [Author Response · Author response to Decision Letter 0]

9 Nov 2023

RESPONSE TO REVIEWERS

1. Is the manuscript technically sound, and do the data support the conclusions?

• Reviewer #1: Yes

• Reviewer #2: Partly

Here, we have reviewed the manuscript for the soundness and ensured that the data supports our conclusions 

2. Has the statistical analysis been performed appropriately and rigorously?

• Reviewer #1: Yes

• Reviewer #2: No 

We have taken note of Reviewer #2’s comment and have revised our statistical analysis to add to the rigor of our methods and results. 

3. Have the authors made all data underlying the findings in their manuscript fully available?

Reviewer #1: Yes

Reviewer #2: No

Reviewer #2, Thank you for your observation regarding the availability of the data underlying our manuscript's findings. Our data was included as Supplementary Data S1. 

4. Is the manuscript presented in an intelligible fashion and written in standard English?

Reviewer #1: Yes

Reviewer #2: No

We took heed of Reviewer #2’s comment and have revised our manuscript for clarity and conciseness. 

5. Review Comments to the Author

Dear Reviewer #1, 

Thank you for your thoughtful and detailed feedback on our manuscript. We sincerely appreciate the time and effort you dedicated to reviewing our work. We have carefully considered each of your points, and your insights have been invaluable in enhancing the quality and clarity of our research.

Please find our responses below.

Reviewer #1: 

Abstract:

• Their objective does not center on exploring the KAP. The term "evaluating" suggests an assessment of interventions, treatments, or comparisons between pre and post studies.

We acknowledge your concern about the term "evaluating" and have revised it to "assessing" to better align with our objective. Thank you for pointing this out.

• They utilized multivariate logistic regression to adjust for other variables and estimate the adjusted odds ratio (AOR) associated with a 95% confidence interval. It might be better to incorporate these details in the methods section.

We have since revised our data analysis plan and have included it in the abstract as “Ordinal logistic regression”.

• Instead of presenting p-values as "p-value <0.001," it would be clearer to state them as "p=xxxe-10." 

Your suggestion regarding the presentation of p-values has been duly noted and implemented. We now express them fully for better clarity.

• Could you provide information on the number of healthcare workers with good/bad practices?

We appreciate your observation regarding the need for information on healthcare workers with good/bad practices. We have revised the abstract to include this important detail.

• In conclusion, the authors did not mention the associated factors that need improvement for raising awareness.

We apologize for the oversight regarding the associated factors for raising awareness. Your feedback has prompted us to revise our conclusion, incorporating recommendations in line with the study findings. 

Methods:

• The authors briefly mention the study design, but a more detailed explanation is necessary.

Thank you for pointing out the need for a more detailed explanation of our study design. We have provided additional information to ensure a comprehensive understanding.

• The sample size calculation provided is quite general and lacks important information, such as the proportion (p) of healthcare workers in the study. A 50% proportion is not appropriate for this type of study. Based on previous research, the authors should calculate the sample size specifically for healthcare workers (HCWs).

Your concerns about the sample size calculation have been addressed. We have recalculated the sample size specifically for healthcare workers based on relevant literature and clarified our approach for achieving a representative sample, as indicated in the next comment. 

• It is stated that this is an online survey, and the authors mention that the survey was conducted until they reached a sample size of 385 HCWs. However, it appears that this survey did not take the sample proportionally, and further clarification is needed.

We revised the sample size calculation based on a Ugandan study by Olum et al. (from which the questionnaire was adopted), where the proportion of sufficient KAP was 69%, 21% and 74%, respectively. We used these KAP scores to calculate the minimum required sample sizes for each outcome variable. We assumed a 95% confidence level (corresponding to a Z-score of 1.96) and a margin of error E of 5% (expressed as 0.05).

1. Knowledge: n = 323 HCWs.

2. Attitudes: n = 257 HCWs.

3. Practices: n = 307 HCWs.

To further enhance the representativeness of the sample, the weights of the specific cadres were targeted to mirror the population, thus, caregivers were to constitute 78% of the sample, administrative staff approximately 12.1%, and environmental health health 9.7% of the sample size. 

In our study, 438 HCWs responded. 

Results:

• In lines 146-147, it would be helpful to provide the number of indicators and the total score of knowledge for better clarification.

We have revised the discussion to provide the number of indicators and the total score of knowledge for better clarification, as suggested.

• In line 189, it is not possible for the 95% confidence interval of the odds ratio (OR) to be negative (-0.05 to 0.15).

Your observation about the negative 95% confidence interval of the odds ratio has been rectified. We have since revised our data analysis to ensure accurate reporting of the results and ensure none of the reported ORs have a negative sign. 

• The study suggests an inverse association between practices and attitudes with knowledge. Is this interpretation accurate?

Regarding the interpretation of the inverse association between practices and attitudes with knowledge, we have reviewed the findings and confirmed the accuracy of our interpretation. This has been elaborated upon in the discussion section. 

Discussion:

• In lines 241-244, the authors suggest that there were more gaps in knowledge and practices among HCWs. They found that HCWs did not exhibit appropriate practices, yet they had sufficient knowledge. However, their data analysis did not support this observation, as it showed that HCWs with insufficient knowledge had good practices. These results appear contradictory.

We appreciate your feedback on the potential contradiction in our interpretation of knowledge and practices among HCWs. We have revisited our discussion and made necessary adjustments to align with the study's results more accurately.

• In line 277, a major limitation of the study was its cross-sectional, which prevented the authors from conducting a more in-depth exploration of the KAP items.

Your point about the limitation of the study being cross-sectional has been duly noted and included in our limitations section.

• While the authors presented a well-written conclusion, it would be beneficial to provide further clarification on the factors that need improvement and how to address them.

Thank you for highlighting the need for further clarification on the factors requiring improvement and how to address them. We have revisited our conclusions and provided additional details to address this concern. 

Once again, we are grateful for your thorough review and constructive feedback. Your input has significantly strengthened the quality and cohesiveness of our manuscript.

 

Dear Reviewer #2,

We appreciate your thoughtful feedback and valuable suggestions regarding our manuscript. Your insights have been extremely instrumental in improving the clarity and quality of our work. We have carefully considered each of your points and made the necessary revisions to address your concerns. We sincerely thank you for your time in reviewing this paper as it is our belief that your comments have greatly elevated the quality of our work. 

Please find our responses below. 

Reviewer #2: 

Title

The title is not easy to understand for the readers

We understand your concern about the readability of the title. Based on your feedback, we have revised the title to make it more straightforward and accessible to readers: "Knowledge, Attitudes, and Practices towards COVID-19 among Healthcare Workers: A Cross-Sectional Survey from Kiambu County, Kenya." We hope this new title is more comprehensible.

Abstract

• Background is well written

Thank you for this encouraging comment.

• The study period (May to August) is not consistent with the mentioned period in the study design on page 4. The questionnaire was used to collect data not for determining associated factors of KAP. Simple and multiple logistic regression, not multivariate logistic regression. See also comments in the methods and materials section. 

We appreciate your attention to detail regarding the study period and the use of the questionnaire. We have made the necessary adjustments to accurately reflect the study period in all instances as 11th March 2021 to 12th August 2021. Thank you for the astute correction regarding the questionnaire and data analysis, we have revised the statement to read “A structured questionnaire was used to explore the factors associated with COVID-19-related KAP”. We have since revised our approach to data analysis as ordinal logistic regression and we have indicated the same in the abstract. 

• Results should be revised according to the comments raised in the methods and materials sections

We have revised the results section in alignment with the updated data and methodology.

• ‘Messenger’ word should be rephrased; authors used this term in the whole manuscript. 

Your feedback on the use of the term "messenger" has been duly noted, and we have rephrased it throughout the manuscript for consistency and clarity. 

Keyword

• Most of the keywords are overlapped with the title of this study

We understand your concern about the overlap between keywords and the title. To address this, we have revised the keywords to ensure they are distinct from the title: Knowledge, attitudes, practices; Healthcare workers; pandemic; Health communication; efficacy beliefs. 

Introduction

• The introduction section is not shaped and written as required for scientific research

We appreciate your feedback on the introduction section. We have restructured and rewritten the introduction to meet the scientific research standards.

•The authors did not consider quality points to highlight why they conducted this study, and about research gaps

The revised introduction now includes quality points highlighting the reasons for conducting the study. 

•Need a comprehensive review of relevant literature

The revised introduction now includes a comprehensive review of relevant literature. Thank you. 

• The objective of this study is not clearly mentioned.

We revised the objective of the study at the end of the Introduction section “This study sought to determine the KAP of HCW towards COVID-19 in Kiambu county, Kenya.”. 

•What are the implications of the findings of this study

The revised introduction now includes a paragraph in the introduction section to cover this, as below: 

“However, given the paramount role of HCWs in pandemic response, what they know, feel and do becomes crucial in the face of a public health emergency. This situation is further complicated by the “infodemic” that paralleled the COVID-19 pandemic [15], and by a supply chain crisis that affected the availability of PPEs for adhering to proper protocols [16]. To avert the higher rate of being infected among HCWs, equipping them with good knowledge and practice is imperative, especially in countries with already low health worker-to-population ratio [17]. Thus, the KAP of HCW towards COVID-19 were critical to the success of the overall COVID-19 response.”

Methods and Materials

• The definition of the study population is not clear. It is also not clear, who are actually HCWs in this study or the way authors defined HCW, required quality modification. 

The above two observations have been noted, and thus the definitions have been revised according to the categories used in our study. In this study, a HCW included any person involved in the provision of health services to a user or those who are on facility grounds employed by the facility. At the time of the study, the county of Kiambu was served by a total of 3700 HCWs. These HCWs were divided into caregivers, administrative staff and environmental health workers, and details of each category can be found below. 

1. Caregivers included all HCWs who interact with patients directly. They included: medical officers, consultants, nurses, clinical officers, dental officers, dental technologists, pharmacists, pharmaceutical staff, laboratory staff, orthopedic technologist, nutritionists, radiographers, physiotherapists and mortuary attendants. 

2. Administrative staff included HCWs who do not interact with patients directly. They included: health administrative officers and staff, health-supportive staff, medical engineering technologists, health records & information officers, medical social workers, ambulance drivers, and HIV testing services staff.

3. Environmental health workers consisted of public health staff, such as community health volunteers, health promotion officers, and public health officers/community health officers

• According to the nature of the study setting and study population, the sample size determination and sampling technique are inappropriate. The sampling on a volunteer basis is not scientifically robust.

In this study, we employed voluntary purposive sampling. While this approach would not be the best, we had to specifically target HCWs who then had to consent to participate in the study. This approach has been used previously in similar KAP studies, such as one by Mark et al., 2022, and was adopted here. 

• It is confusing, what type of questionnaire the authors used to collect the data: structured/open-ended/semi-structured? How could a questionnaire be structured and open-ended simultaneously? Probably, the authors considered a semi-structured research tool

Thank you for this observation. It is correct that we employed a semi-structured questionnaire to collect data, but this manuscript only details the results of the structured questionnaire. We have corrected the manuscript accordingly. 

• The authors did not mention how they checked the reliability, validity, and consistency of their data and questionnaire

This is a good observation and has been addressed in the manuscript in the section headed as “Questionnaire”. The main additions can be found below:

We employed a questionnaire from Olum et al.(Olum et al., 2020) for this study. This questionnaire was constructed as such: 

1. Knowledge was assessed using a 11-item questionnaire adapted from Zhong et al., and modified to suit HCWs, each correct answer weighing one point. According to Zhong et al., this was a reliable knowledge scale for adoption as it possessed a Cronbach’s alpha score of 0.71 (Zhong et al., 2020). 

 We adopted this Knowledge scale for our study, however we added three context-specific questions regarding the location of Kiambu county’s COVID-19 isolaton centre, handling a suspected COVID-19 case and data entry for suspected COVID-19 cases. These questions were selected and reviewed by public health experts. 

2. Attitudes were assessed using 5 Likert-item questions that have been adopted from Goni et al. and modified by Olum et al. Goni et al.’s Cronbach’s alpha scale measurement for this was 0.77 (Goni et al., 2020).

We adopted this Attitude scale for our study, however we added six context-specific questions regarding how the national and county government structures were handling the pandemic at various stages.

3. Practices were assessed using five Likert-item questions that have been developed from the WHO recommended practices. This was adopted fully from the Olum et al.

The alpha measurements for the scale subsets as used in our study were as follows: αk = 0.69, αa = 0.81 and αp = 0.61. In this study, a Cronbach’s alpha score of above 0.6 was considered adequate, similar to other KAP studies (Gopalakrishnan et al., 2021; Majmundar et al., 2018; Mark et al., 2022; Sayili et al., 2022). 

• The authors did not mention what was the basis of the classification of the target variables

This is duly noted. We have revised our study to use Bloom’s cutoff points as the basis of classifying our target variables into Good (>80%), Medium (60-79%) and Poor (<60%). 

• The authors did not check the bivariate association of the target variables with the predictor variables, so, how did they select probable predictor variables for multiple regression modeling? In the modeling, some potential predictors may be insignificant due to arbitrary selection of the predictors. 

Given this comment, we had revisited the analysis and revised the data analysis plan to conduct ordinal logistic regression given the application of Bloom’s cutoff points that divided each KAP target variable into Good, Medium and Poor. 

• The authors actually performed binary and multiple logistic regression but they mentioned “multivariate logistic regression”

This is duly noted and has been modified according to our new analysis. Thank you. 

• How the authors selected the link function for logistic regression as they did not check the nature of the target variable

Thank you for this observation. We have revisited our analyses and reorganized our target variable according to the Bloom’s cutoff points for KAP studies and have organized our outcomes, such as type of knowledge into: Good, Medium and Poor. The link function was chosen based on the changes in the cumulative probabilities: the “probit” link function for gradual changes (as noted with the knowledge and attitude types) and the complementary log-log link function for practice, for which the cumulative probabilities increased from 0 slowly and then rapidly approached (Chapter 12 Ordinal Logistic Regression | Companion to BER 642: Advanced Regression Methods, n.d.).

Results

• The results should be revised according to the comments in the methodology section

This was duly noted and the entirety of the results section has been revised thoroughly. 

Authors should also consider the following specific observations:

• The results are poorly written and presented

Thank you for this observation. We have rewritten the entirety of the results section again for enhanced clarity. 

 - We have thoroughly reviewed and refined the presentation of our results. The results of the ordinal logistic regression now include all the KAP outcomes and only the significant variables. This approach enhances the clarity of our findings and makes the presentation more precise and accessible to readers.

We hope these revisions address your concerns about the clarity and presentation of our results. 

• Tables are too long

Thank you for your observations regarding the presentation of the results. We greatly value your feedback and have taken your comments into serious consideration. Based on your suggestions, we have made the following revisions to address your concerns:

1. Tables Length:

 - We acknowledge (and share) your concern about the length of the tables. To address this, we have revised Table 1 to include only the socio-demographic characteristics of the sample. Cadre proportions have been mentioned in the narrative section of the results, ensuring a more concise presentation of information. We have also included the data pertaining to the specific cadres as Supplementary material S3. 

 - Additionally, we have truncated the other tables where possible, ensuring a more focused and streamlined presentation of the data.

- We have also included the crude odds ratio results and the p-values as Supplementary material S2. 

• Authors unnecessarily mentioned too long questions in the text

We have revised the results section to eliminate references to long questions, ensuring a more concise and reader-friendly presentation of our findings.

• Authors used ‘n’ for indicating sample size in the methodology section; in the result section it is confusing about the notation ‘n’ and ‘N’

We have rectified the use of 'n' and 'N' to accurately represent whether it refers to the sample or population size, ensuring clarity in both the methodology and results sections.

• In Tables 4, 6, and 8, the authors provided a column ‘p-value’. P-value of which estimates: crude odds ratio (COR) or adjusted odds ratio (AOR)? If it indicates AOR, then where is the significance value of the COR estimates? Without significance value, how these results could be interpretable?

We thank you again, for this invaluable comment and apologize for its omission. In the revised manuscript, we have included the coefficient, AOR (with confidence intervals in brackets) and p-values only.

• It is clear from Tables 4, 6, and 8, that the authors performed simple logistic regression for all predictor variables as well as performed multiple logistic regression including all the predictor variables in the model, Why? Why did the authors consider the predictors with insignificant COR in the multiple logistic regression modeling?

We thank you again, for this invaluable comment. We have revised the manuscript according to the refined results of ordinal logistic regression. For this analysis, we only included predictors with COR with p-values less than < 0.05 in our initial and final fitted models. 

• Estimates and p-values are not presented in a unique and standard format.

The estimates and p-values in our results section have been restructured to adhere to a standard format, enhancing the consistency and readability of the presentation.

• The second column of Tables 4, 6, and 8 should be renamed as ‘Label’

Due to the restructuring of our analysis, the specific tables you referred to no longer exist. Nevertheless, we appreciate your suggestion, which has guided our approach in creating clearer and more organized tables. We are grateful for the direction. 

• In Table 4, the authors provided COR for attitude and practice considering these as predictor variables of knowledge. In principle, how attitude and practice could be the predictors of knowledge? Is there any basis or reference against this issue? If yes, why there is no adjusted odds ratio for these factors? Probably, the authors considered it wrongly. Authors should rethink and redesign

We have since rethought and redesigned our analysis of our data to ordinal logistic regression. We have only considered the independent variables as predictors for each KAP dependent variable. These variables were chosen only if they had a crude odds ratio COR of less than p < 0.05. 

• In Tables 6 and 8, the authors did not consider the knowledge level as a predictor variable for attitude and practice, why?

We have since rethought and redesigned our analysis of our data to ordinal logistic regression. This approach ensures a more accurate representation of the relationship between predictor variables and outcomes, addressing the concerns you raised regarding the inclusion of attitude, practice, and knowledge levels as predictors.

• Is the result of correlation measurement between knowledge and practice correct? If yes, does this type of relationship exist in reality? The authors did not discuss the reason for this result in the discussion section. What would be the implications of this finding?

In response to your query, we did conduct Spearman’s rank correlation for the non-normally distributed knowledge scores against the practice scores. The analysis indeed revealed an inverse relationship between knowledge and practice, which was statistically significant. However, we acknowledge that our discussion of this relationship in the manuscript was not as clear and comprehensive as it should have been.

Your feedback has prompted us to reevaluate our approach, and we have decided to restructure the discussion section into distinct segments. Specifically, we will now have separate sections dedicated to discussing the knowledge, attitude, and practice findings individually. Additionally, we will include a dedicated section to explore the relationship between knowledge and practice in greater depth. This restructuring will enable us to thoroughly analyze the implications of this finding and provide a more detailed and nuanced discussion on this significant aspect of our research.

We are genuinely grateful for your astute observation, which has guided us to enhance the clarity and depth of our discussion. 

Discussion

• The discussion section is not written as expected for a scientific article

In response to your comments, we have diligently revised the discussion section to ensure a proper and accurate reflection of our methods and results.

• The authors are suggested to revise the methodology and results according to the comments raised in the respective sections

We have taken time to review both the methods and results section for congruency with our discussion. 

• They should rewrite this section by explaining the potential findings with proper supporting citations

We have taken care to address the concerns raised in the respective sections and have provided a comprehensive explanation of our study findings. Additionally, we have strengthened our discussion by incorporating relevant literature to support our conclusions. 

• Authors are also suggested to add implications of the findings and limitation of the study

To further enhance the quality of our manuscript, we have included a dedicated section discussing the implications of our study findings. Furthermore, we have outlined the limitations of our study, acknowledging the constraints and challenges we encountered during our research.

References quoted: 

Chapter 12 Ordinal Logistic Regression | Companion to BER 642: Advanced Regression Methods. (n.d.). Retrieved November 4, 2023, from https://bookdown.org/chua/ber642_advanced_regression/ordinal-logistic-regression.html

Goni, M. D., Naing, N. N., Hasan, H., Wan-Arfah, N., Deris, Z. Z., Arifin, W. N., Hussin, T. M. A. R., Abdulrahman, A. S., Baaba, A. A., & Arshad, M. R. (2020). Development and validation of knowledge, attitude and practice questionnaire for prevention of respiratory tract infections among Malaysian Hajj pilgrims. BMC Public Health, 20(1), 1–10. https://doi.org/10.1186/S12889-020-8269-9/TABLES/5

Gopalakrishnan, S., Kandasamy, S., Abraham, B., Senthilkumar, M., & Almohammed, O. A. (2021). Knowledge, Attitude, and Practices Associated With COVID-19 Among Healthcare Workers in Hospitals: A Cross-Sectional Study in India. Frontiers in Public Health, 9, 787845. https://doi.org/10.3389/FPUBH.2021.787845

Majmundar, A., Allem, J. P., Cruz, T. B., & Unger, J. B. (2018). The Why We Retweet scale. PLoS ONE, 13(10). https://doi.org/10.1371/journal.pone.0206076

Mark, E., Udod, G., Skinner, J., & Jones, M. (2022a). Knowledge, attitudes, and practices [KAP] toward COVID-19: A cross-sectional study in the New York Metropolitan Area and California Bay Area. PLOS ONE, 17(8), e0271212. https://doi.org/10.1371/JOURNAL.PONE.0271212

Mark, E., Udod, G., Skinner, J., & Jones, M. (2022b). Knowledge, attitudes, and practices [KAP] toward COVID-19: A cross-sectional study in the New York Metropolitan Area and California Bay Area. PLoS ONE, 17(8 August). https://doi.org/10.1371/journal.pone.0271212

Olum, R., Chekwech, G., Wekha, G., Nassozi, D. R., & Bongomin, F. (2020). Coronavirus Disease-2019: Knowledge, Attitude, and Practices of Health Care Workers at Makerere University Teaching Hospitals, Uganda. Frontiers in Public Health, 8. https://doi.org/10.3389/fpubh.2020.00181

Sayili, U., Siddikoglu, E., Pirdal, B. Z., Uygur, A., Toplu, F. S., & Can, G. (2022). The heat wave knowledge, awareness, practice and behavior scale: Scale development, validation and reliability. PLoS ONE, 17(12 December). https://doi.org/10.1371/journal.pone.0279259

Zhong, B. L., Luo, W., Li, H. M., Zhang, Q. Q., Liu, X. G., Li, W. T., & Li, Y. (2020). Knowledge, attitudes, and practices towards COVID-19 among Chinese residents during the rapid rise period of the COVID-19 outbreak: a quick online cross-sectional survey. International Journal of Biological Sciences, 16(10), 1745. https://doi.org/10.7150/IJBS.45221

---

## [Editor Report · Decision Letter 1]

20 Nov 2023

PONE-D-23-19304R1Knowledge, attitudes and practices towards COVID-19 among healthcare workers: a cross-sectional survey from Kiambu County, KenyaPLOS ONE

Dear Dr. Juttla,

Thank you for submitting your manuscript to PLOS ONE. After careful consideration, we feel that it has merit but does not fully meet PLOS ONE’s publication criteria as it currently stands. Therefore, we invite you to submit a revised version of the manuscript that addresses the points raised during the review process.

We look forward to receiving your revised manuscript.

Kind regards,

Probir Kumar Ghosh, MSc.

Guest Editor

PLOS ONE

Additional Editor Comments:

EDITOR’S RESPONSE TO REVIEWERS

The authors should consider editor's comments in the manuscript.

1. Is the manuscript technically sound, and do the data support the conclusions?

• Reviewer #1: Yes

• Reviewer #2: Partly

Here, we have reviewed the manuscript for the soundness and ensured that the data supports our conclusions

Editor’s response: The reviewer’s comments have fully addressed in the manuscript text.

2. Has the statistical analysis been performed appropriately and rigorously?

• Reviewer #1: Yes

• Reviewer #2: No

We have taken note of Reviewer #2’s comment and have revised our statistical analysis to add to the rigor of our methods and results.

Editor’s response: The reviewer’s comments have not yet fully addressed in the manuscript text.

3. Have the authors made all data underlying the findings in their manuscript fully available?

Reviewer #1: Yes

Reviewer #2: No

Reviewer #2, Thank you for your observation regarding the availability of the data underlying our manuscript's findings. Our data was included as Supplementary Data S1.

Editor’s response: The reviewer’s comments have fully addressed in the manuscript text.

4. Is the manuscript presented in an intelligible fashion and written in standard English?

Reviewer #1: Yes

Reviewer #2: No

We took heed of Reviewer #2’s comment and have revised our manuscript for clarity and conciseness.

Editor’s response: The reviewer’s comments have partly addressed in the manuscript text.

5. Review Comments to the Author

Dear Reviewer #1,

Thank you for your thoughtful and detailed feedback on our manuscript. We sincerely appreciate the time and effort you dedicated to reviewing our work. We have carefully considered each of your points, and your insights have been invaluable in enhancing the quality and clarity of our research.

Please find our Editor’s responses below.

Reviewer #1:

Abstract:

• Their objective does not center on exploring the KAP. The term "evaluating" suggests an assessment of interventions, treatments, or comparisons between pre and post studies.

We acknowledge your concern about the term "evaluating" and have revised it to "assessing" to better align with our objective. Thank you for pointing this out.

Editor’s response: The reviewer’s comments have not yet fully addressed in the manuscript text.

• They utilized multivariate logistic regression to adjust for other variables and estimate the adjusted odds ratio (AOR) associated with a 95% confidence interval. It might be better to incorporate these details in the methods section.

We have since revised our data analysis plan and have included it in the abstract as “Ordinal logistic regression”.

Editor’s response: The manuscript still requires further revision to address the reviewer's comments adequately. The authors should consider reanalyzing the data using multivariate regression models.

• Instead of presenting p-values as "p-value <0.001," it would be clearer to state them as "p=xxxe-10."

Your suggestion regarding the presentation of p-values has been duly noted and implemented. We now express them fully for better clarity.

Editor’s response: The reviewer’s comments have not yet fully addressed in the manuscript tables.

• Could you provide information on the number of healthcare workers with good/bad practices?

We appreciate your observation regarding the need for information on healthcare workers with good/bad practices. We have revised the abstract to include this important detail.

Editor’s response: The reviewer’s comments have not yet fully addressed in the manuscript text.

• In conclusion, the authors did not mention the associated factors that need improvement for raising awareness.

We apologize for the oversight regarding the associated factors for raising awareness. Your feedback has prompted us to revise our conclusion, incorporating recommendations in line with the study findings.

Editor’s response: The reviewer’s comments have partly addressed in the manuscript text.

Methods:

• The authors briefly mention the study design, but a more detailed explanation is necessary.

Thank you for pointing out the need for a more detailed explanation of our study design. We have provided additional information to ensure a comprehensive understanding.

Editor’s response: The reviewer’s comments have fully addressed in the manuscript text.

• The sample size calculation provided is quite general and lacks important information, such as the proportion (p) of healthcare workers in the study. A 50% proportion is not appropriate for this type of study. Based on previous research, the authors should calculate the sample size specifically for healthcare workers (HCWs).

• It is stated that this is an online survey, and the authors mention that the survey was conducted until they reached a sample size of 385 HCWs. However, it appears that this survey did not take the sample proportionally, and further clarification is needed.

Your concerns about the sample size calculation have been addressed. We have recalculated the sample size specifically for healthcare workers based on relevant literature and clarified our approach for achieving a representative sample, as below:

We revised the sample size calculation based on a Ugandan study by Olum et al. (from which the questionnaire was adopted), where the proportion of sufficient KAP was 69%, 21% and 74%, respectively. We used these KAP scores to calculate the minimum required sample sizes for each outcome variable. We assumed a 95% confidence level (corresponding to a Z-score of 1.96) and a margin of error E of 5% (expressed as 0.05).

1. Knowledge: n = 323 HCWs.

2. Attitudes: n = 257 HCWs.

3. Practices: n = 307 HCWs.

To further enhance the representativeness of the sample, the weights of the specific cadres were targeted to mirror the population, thus, caregivers were to constitute 78% of the sample, administrative staff approximately 12.1%, and environmental health health 9.7% of the sample size.

In our study, 438 HCWs responded.

Editor’s response: The reviewer’s comments have fully addressed in the manuscript text.

Results:

• In lines 146-147, it would be helpful to provide the number of indicators and the total score of knowledge for better clarification.

We have revised the discussion to provide the number of indicators and the total score of knowledge for better clarification, as suggested.

Editor’s response: The reviewer’s comments have fully addressed in the manuscript text.

• In line 189, it is not possible for the 95% confidence interval of the odds ratio (OR) to be negative (-0.05 to 0.15).

Your observation about the negative 95% confidence interval of the odds ratio has been rectified. We have since revised our data analysis to ensure accurate reporting of the results and ensure none of the reported ORs have a negative sign.

Editor’s response: The reviewer’s comments have fully addressed in the manuscript text.

• The study suggests an inverse association between practices and attitudes with knowledge. Is this interpretation accurate?

Regarding the interpretation of the inverse association between practices and attitudes with knowledge, we have reviewed the findings and confirmed the accuracy of our interpretation. This has been elaborated upon in the discussion section.

Editor’s response: The reviewer’s comments have not yet fully addressed in the manuscript text. The authors should require reanalysis the data using statistical models and interpretation properly.

Discussion:

• In lines 241-244, the authors suggest that there were more gaps in knowledge and practices among HCWs. They found that HCWs did not exhibit appropriate practices, yet they had sufficient knowledge. However, their data analysis did not support this observation, as it showed that HCWs with insufficient knowledge had good practices. These results appear contradictory.

We appreciate your feedback on the potential contradiction in our interpretation of knowledge and practices among HCWs. We have revisited our discussion and made necessary adjustments to align with the study's results more accurately.

Editor’s response: The reviewer’s comments have not yet fully addressed in the manuscript text.

• In line 277, a major limitation of the study was its cross-sectional, which prevented the authors from conducting a more in-depth exploration of the KAP items.

Your point about the limitation of the study being cross-sectional has been duly noted and included in our limitations section.

Editor’s response: The reviewer’s comments have partly addressed in the manuscript text.

• While the authors presented a well-written conclusion, it would be beneficial to provide further clarification on the factors that need improvement and how to address them.

Thank you for highlighting the need for further clarification on the factors requiring improvement and how to address them. We have revisited our conclusions and provided additional details to address this concern.

Once again, we are grateful for your thorough review and constructive feedback. Your input has significantly strengthened the quality and cohesiveness of our manuscript.

 

Dear Reviewer #2,

We appreciate your thoughtful feedback and valuable suggestions regarding our manuscript. Your insights have been extremely instrumental in improving the clarity and quality of our work. We have carefully considered each of your points and made the necessary revisions to address your concerns. We sincerely thank you for your time in reviewing this paper as it is our belief that your comments have greatly elevated the quality of our work.

Please find our Editor’s responses below.

Reviewer #2:

Title

The title is not easy to understand for the readers

We understand your concern about the readability of the title. Based on your feedback, we have revised the title to make it more straightforward and accessible to readers: "Knowledge, Attitudes, and Practices towards COVID-19 among Healthcare Workers: A Cross-Sectional Survey from Kiambu County, Kenya." We hope this new title is more comprehensible.

Editor’s response: The reviewer’s comments have fully addressed in the manuscript text.

Abstract

• Background is well written

Thank you for this encouraging comment.

• The study period (May to August) is not consistent with the mentioned period in the study design on page 4. The questionnaire was used to collect data not for determining associated factors of KAP. Simple and multiple logistic regression, not multivariate logistic regression. See also comments in the methods and materials section.

We appreciate your attention to detail regarding the study period and the use of the questionnaire. We have made the necessary adjustments to accurately reflect the study period in all instances as 11th March 2021 to 12th August 2021. Thank you for the astute correction regarding the questionnaire and data analysis, we have revised the statement to read “A structured questionnaire was used to explore the factors associated with COVID-19-related KAP”. We have since revised our approach to data analysis as ordinal logistic regression and we have indicated the same in the abstract.

Editor’s response: The manuscript still requires further revision to address the reviewer's comments adequately. The authors should consider reanalyzing the data using multivariate regression models.

• Results should be revised according to the comments raised in the methods and materials sections

We have revised the results section in alignment with the updated data and methodology.

• ‘Messenger’ word should be rephrased; authors used this term in the whole manuscript.

Your feedback on the use of the term "messenger" has been duly noted, and we have rephrased it throughout the manuscript for consistency and clarity.

Editor’s response: The reviewer’s comments have fully addressed in the manuscript text.

Keyword

• Most of the keywords are overlapped with the title of this study

We understand your concern about the overlap between keywords and the title. To address this, we have revised the keywords to ensure they are distinct from the title: Knowledge, attitudes, practices; Healthcare workers; pandemic; Health communication; efficacy beliefs.

Editor’s response: The reviewer’s comments have fully addressed in the manuscript text.

Introduction

• The introduction section is not shaped and written as required for scientific research

We appreciate your feedback on the introduction section. We have restructured and rewritten the introduction to meet the scientific research standards.

Editor’s response: The reviewer’s comments have not yet fully addressed in the manuscript text.

•The authors did not consider quality points to highlight why they conducted this study, and about research gaps

The revised introduction now includes quality points highlighting the reasons for conducting the study.

Editor’s response: The reviewer’s comments have fully addressed in the manuscript text.

•Need a comprehensive review of relevant literature

The revised introduction now includes a comprehensive review of relevant literature. Thank you.

Editor’s response: The reviewer’s comments have fully addressed in the manuscript text.

• The objective of this study is not clearly mentioned.

We revised the objective of the study at the end of the Introduction section “This study sought to determine the KAP of HCW towards COVID-19 in Kiambu county, Kenya.”.

Editor’s response: The reviewer’s comments have not fully addressed in the manuscript text.

•What are the implications of the findings of this study

The revised introduction now includes a paragraph in the introduction section to cover this, as below:

“However, given the paramount role of HCWs in pandemic Editor’s response, what they know, feel and do becomes crucial in the face of a public health emergency. This situation is further complicated by the “infodemic” that paralleled the COVID-19 pandemic [15], and by a supply chain crisis that affected the availability of PPEs for adhering to proper protocols [16]. To avert the higher rate of being infected among HCWs, equipping them with good knowledge and practice is imperative, especially in countries with already low health worker-to-population ratio [17]. Thus, the KAP of HCW towards COVID-19 were critical to the success of the overall COVID-19 Editor’s response.”

Methods and Materials

• The definition of the study population is not clear. It is also not clear, who are actually HCWs in this study or the way authors defined HCW, required quality modification.

The above two observations have been noted, and thus the definitions have been revised according to the categories used in our study. In this study, a HCW included any person involved in the provision of health services to a user or those who are on facility grounds employed by the facility. At the time of the study, the county of Kiambu was served by a total of 3700 HCWs. These HCWs were divided into caregivers, administrative staff and environmental health workers, and details of each category can be found below.

1. Caregivers included all HCWs who interact with patients directly. They included: medical officers, consultants, nurses, clinical officers, dental officers, dental technologists, pharmacists, pharmaceutical staff, laboratory staff, orthopedic technologist, nutritionists, radiographers, physiotherapists and mortuary attendants.

2. Administrative staff included HCWs who do not interact with patients directly. They included: health administrative officers and staff, health-supportive staff, medical engineering technologists, health records & information officers, medical social workers, ambulance drivers, and HIV testing services staff.

3. Environmental health workers consisted of public health staff, such as community health volunteers, health promotion officers, and public health officers/community health officers

Editor’s response: The reviewer’s comments have not fully addressed in the manuscript text.

• According to the nature of the study setting and study population, the sample size determination and sampling technique are inappropriate. The sampling on a volunteer basis is not scientifically robust.

In this study, we employed voluntary purposive sampling. While this approach would not be the best, we had to specifically target HCWs who then had to consent to participate in the study. This approach has been used previously in similar KAP studies, such as one by Mark et al., 2022, and was adopted here.

Editor’s response: The reviewer’s comments have fully addressed in the manuscript text.

• It is confusing, what type of questionnaire the authors used to collect the data: structured/open-ended/semi-structured? How could a questionnaire be structured and open-ended simultaneously? Probably, the authors considered a semi-structured research tool

Thank you for this observation. It is correct that we employed a semi-structured questionnaire to collect data, but this manuscript only details the results of the structured questionnaire. We have corrected the manuscript accordingly.

Editor’s response: The reviewer’s comments have fully addressed in the manuscript text.

• The authors did not mention how they checked the reliability, validity, and consistency of their data and questionnaire

This is a good observation and has been addressed in the manuscript in the section headed as “Questionnaire”. The main additions can be found below:

We employed a questionnaire from Olum et al.(Olum et al., 2020) for this study. This questionnaire was constructed as such:

1. Knowledge was assessed using a 11-item questionnaire adapted from Zhong et al., and modified to suit HCWs, each correct answer weighing one point. According to Zhong et al., this was a reliable knowledge scale for adoption as it possessed a Cronbach’s alpha score of 0.71 (Zhong et al., 2020).

We adopted this Knowledge scale for our study, however we added three context-specific questions regarding the location of Kiambu county’s COVID-19 isolaton centre, handling a suspected COVID-19 case and data entry for suspected COVID-19 cases. These questions were selected and reviewed by public health experts.

2. Attitudes were assessed using 5 Likert-item questions that have been adopted from Goni et al. and modified by Olum et al. Goni et al.’s Cronbach’s alpha scale measurement for this was 0.77 (Goni et al., 2020).

We adopted this Attitude scale for our study, however we added six context-specific questions regarding how the national and county government structures were handling the pandemic at various stages.

3. Practices were assessed using five Likert-item questions that have been developed from the WHO recommended practices. This was adopted fully from the Olum et al.

The alpha measurements for the scale subsets as used in our study were as follows: αk = 0.69, αa = 0.81 and αp = 0.61. In this study, a Cronbach’s alpha score of above 0.6 was considered adequate, similar to other KAP studies (Gopalakrishnan et al., 2021; Majmundar et al., 2018; Mark et al., 2022; Sayili et al., 2022).

Editor’s response: The reviewer’s comments have fully addressed in the manuscript text.

• The authors did not mention what was the basis of the classification of the target variables

This is duly noted. We have revised our study to use Bloom’s cutoff points as the basis of classifying our target variables into Good (>80%), Medium (60-79%) and Poor (<60%).

Editor’s response: The reviewer’s comments have fully addressed in the manuscript text.

• The authors did not check the bivariate association of the target variables with the predictor variables, so, how did they select probable predictor variables for multiple regression modeling? In the modeling, some potential predictors may be insignificant due to arbitrary selection of the predictors.

Given this comment, we had revisited the analysis and revised the data analysis plan to conduct ordinal logistic regression given the application of Bloom’s cutoff points that divided each KAP target variable into Good, Medium and Poor.

Editor’s response: The reviewer’s comments have fully addressed in the manuscript text.

• The authors actually performed binary and multiple logistic regression but they mentioned “multivariate logistic regression”

This is duly noted and has been modified according to our new analysis. Thank you.

Editor’s response: The reviewer’s comments have not yet fully addressed in the manuscript text.

• How the authors selected the link function for logistic regression as they did not check the nature of the target variable

Thank you for this observation. We have revisited our analyses and reorganized our target variable according to the Bloom’s cutoff points for KAP studies and have organized our outcomes, such as type of knowledge into: Good, Medium and Poor. The link function was chosen based on the changes in the cumulative probabilities: the “probit” link function for gradual changes (as noted with the knowledge and attitude types) and the complementary log-log link function for practice, for which the cumulative probabilities increased from 0 slowly and then rapidly approached (Chapter 12 Ordinal Logistic Regression | Companion to BER 642: Advanced Regression Methods, n.d.).

Editor’s response: The reviewer’s comments have not yet fully addressed in the manuscript text.

Results

• The results should be revised according to the comments in the methodology section

This was duly noted and the entirety of the results section has been revised thoroughly.

Editor’s response: The reviewer’s comments have fully addressed in the manuscript text.

Authors should also consider the following specific observations:

• The results are poorly written and presented

Thank you for this observation. We have rewritten the entirety of the results section again for enhanced clarity.

- We have thoroughly reviewed and refined the presentation of our results. The results of the ordinal logistic regression now include all the KAP outcomes and only the significant variables. This approach enhances the clarity of our findings and makes the presentation more precise and accessible to readers.

We hope these revisions address your concerns about the clarity and presentation of our results.

Editor’s response: The reviewer’s comments have partly addressed in the manuscript text.

• Tables are too long

Thank you for your observations regarding the presentation of the results. We greatly value your feedback and have taken your comments into serious consideration. Based on your suggestions, we have made the following revisions to address your concerns:

1. Tables Length:

- We acknowledge (and share) your concern about the length of the tables. To address this, we have revised Table 1 to include only the socio-demographic characteristics of the sample. Cadre proportions have been mentioned in the narrative section of the results, ensuring a more concise presentation of information. We have also included the data pertaining to the specific cadres as Supplementary material S3.

- Additionally, we have truncated the other tables where possible, ensuring a more focused and streamlined presentation of the data.

- We have also included the crude odds ratio results and the p-values as Supplementary material S2.

Editor’s response: The reviewer’s comments have not fully addressed in the manuscript text. Instead the authors may create graphs for presenting the data.

• Authors unnecessarily mentioned too long questions in the text

We have revised the results section to eliminate references to long questions, ensuring a more concise and reader-friendly presentation of our findings.

Editor’s response: The reviewer’s comments have partly addressed in the manuscript text.

• Authors used ‘n’ for indicating sample size in the methodology section; in the result section it is confusing about the notation ‘n’ and ‘N’

We have rectified the use of 'n' and 'N' to accurately represent whether it refers to the sample or population size, ensuring clarity in both the methodology and results sections.

Editor’s response: The reviewer’s comments have fully addressed in the manuscript text.

• In Tables 4, 6, and 8, the authors provided a column ‘p-value’. P-value of which estimates: crude odds ratio (COR) or adjusted odds ratio (AOR)? If it indicates AOR, then where is the significance value of the COR estimates? Without significance value, how these results could be interpretable?

We thank you again, for this invaluable comment and apologize for its omission. In the revised manuscript, we have included the coefficient, AOR (with confidence intervals in brackets) and p-values only.

Editor’s response: The reviewer’s comments have not yet fully addressed in the manuscript text.

• It is clear from Tables 4, 6, and 8, that the authors performed simple logistic regression for all predictor variables as well as performed multiple logistic regression including all the predictor variables in the model, Why? Why did the authors consider the predictors with insignificant COR in the multiple logistic regression modeling?

We thank you again, for this invaluable comment. We have revised the manuscript according to the refined results of ordinal logistic regression. For this analysis, we only included predictors with COR with p-values less than < 0.05 in our initial and final fitted models.

Editor’s response: The reviewer’s comments have not fully addressed in the manuscript text. The authors should consider multivariate associated models instead of predictive models.

• Estimates and p-values are not presented in a unique and standard format.

The estimates and p-values in our results section have been restructured to adhere to a standard format, enhancing the consistency and readability of the presentation.

Editor’s response: The reviewer’s comments have fully addressed in the manuscript text.

• The second column of Tables 4, 6, and 8 should be renamed as ‘Label’

Due to the restructuring of our analysis, the specific tables you referred to no longer exist. Nevertheless, we appreciate your suggestion, which has guided our approach in creating clearer and more organized tables. We are grateful for the direction.

Editor’s response: The reviewer’s comments have fully addressed in the manuscript text.

• In Table 4, the authors provided COR for attitude and practice considering these as predictor variables of knowledge. In principle, how attitude and practice could be the predictors of knowledge? Is there any basis or reference against this issue? If yes, why there is no adjusted odds ratio for these factors? Probably, the authors considered it wrongly. Authors should rethink and redesign

We have since rethought and redesigned our analysis of our data to ordinal logistic regression. We have only considered the independent variables as predictors for each KAP dependent variable. These variables were chosen only if they had a crude odds ratio COR of less than p < 0.05.

Editor’s response: same as before. The author should require reanalysis data using associated regression models instead of predictive models.

• In Tables 6 and 8, the authors did not consider the knowledge level as a predictor variable for attitude and practice, why?

We have since rethought and redesigned our analysis of our data to ordinal logistic regression. This approach ensures a more accurate representation of the relationship between predictor variables and outcomes, addressing the concerns you raised regarding the inclusion of attitude, practice, and knowledge levels as predictors.

Editor’s response: The reviewer’s comments have fully addressed in the manuscript text.

• Is the result of correlation measurement between knowledge and practice correct? If yes, does this type of relationship exist in reality? The authors did not discuss the reason for this result in the discussion section. What would be the implications of this finding?

In Editor’s response to your query, we did conduct Spearman’s rank correlation for the non-normally distributed knowledge scores against the practice scores. The analysis indeed revealed an inverse relationship between knowledge and practice, which was statistically significant. However, we acknowledge that our discussion of this relationship in the manuscript was not as clear and comprehensive as it should have been.

Your feedback has prompted us to reevaluate our approach, and we have decided to restructure the discussion section into distinct segments. Specifically, we will now have separate sections dedicated to discussing the knowledge, attitude, and practice findings individually. Additionally, we will include a dedicated section to explore the relationship between knowledge and practice in greater depth. This restructuring will enable us to thoroughly analyze the implications of this finding and provide a more detailed and nuanced discussion on this significant aspect of our research.

We are genuinely grateful for your astute observation, which has guided us to enhance the clarity and depth of our discussion.

Editor’s response: The reviewer’s comments have fully addressed in the manuscript text.

Discussion

• The discussion section is not written as expected for a scientific article

In Editor’s response to your comments, we have diligently revised the discussion section to ensure a proper and accurate reflection of our methods and results.

Editor’s response: The reviewer’s comments have partly addressed in the manuscript text.

• The authors are suggested to revise the methodology and results according to the comments raised in the respective sections

We have taken time to review both the methods and results section for congruency with our discussion.

Editor’s response: The reviewer’s comments have fully addressed in the manuscript text.

• They should rewrite this section by explaining the potential findings with proper supporting citations

We have taken care to address the concerns raised in the respective sections and have provided a comprehensive explanation of our study findings. Additionally, we have strengthened our discussion by incorporating relevant literature to support our conclusions.

Editor’s response: The reviewer’s comments have fully addressed in the manuscript text.

• Authors are also suggested to add implications of the findings and limitation of the study

To further enhance the quality of our manuscript, we have included a dedicated section discussing the implications of our study findings. Furthermore, we have outlined the limitations of our study, acknowledging the constraints and challenges we encountered during our research.

Editor’s response: The reviewer’s comments have fully addressed in the manuscript text.

---

## [Author Response · Author response to Decision Letter 1]

29 Dec 2023

AUTHOR’S RESPONSE

EDITOR’S RESPONSE TO AUTHORS

1. Is the manuscript technically sound, and do the data support the conclusions?

• Reviewer #1: Yes

• Reviewer #2: Partly

Here, we have reviewed the manuscript for the soundness and ensured that the data supports our conclusions 

Editor’s response: The reviewer’s comments have fully addressed in the manuscript text.

Author’s response: Nothing to address

2. Has the statistical analysis been performed appropriately and rigorously?

• Reviewer #1: Yes

• Reviewer #2: No 

We have taken note of Reviewer #2’s comment and have revised our statistical analysis to add to the rigor of our methods and results. 

Editor’s response: The reviewer’s comments have not yet fully addressed in the manuscript text.

Author’s response: We thank the Academic Editor for their insight regarding the data analysis and interpretation. However, before we re-analyze our data again, we would like to offer our perspective on the statistical analysis we performed and ask for further clarification for the same. 

Initially, our outcome variable was BINARY, that is, either good or bad Knowledge, Attitudes or Practices based on the scores being either above (good) or below (bad) the mean score for KAP. For this, we calculated the crude and adjusted odds ratios (bivariate and multivariable logistic regression). For our multivariable analysis, we modelled both biologically plausible (age, sex, cadre) and the significant variables (bivariate analysis p < 0.05) into the model. 

However, Reviewer 2 questioned this approach and questioned the variables in the final multivariable models. They advised us to “rethink and redesign” our model and consider only independent variables with a p-value < 0.05 for the multivariable analysis. 

We agreed with this observation and decided to adjust our outcome variable with a standard criteria used by several KAP studies, which is the Bloom’s cutoff points (Good >80%; Medium 60-79%; and Poor <60%). This classified our outcome variable into 3 ordered categories: Good, Medium and Poor. 

Given that our 3 dependent variables are now ordinal (good knowledge, good attitude and good practice), we chose to model our data using multivariable ordinal logistic regression. Ordinal logistic regression is a statistical analysis method that can be used to model the relationship between an ordinal response variable and one or more explanatory variables. Further, an ordinal variable is a categorical variable for which there is a clear ordering of the category levels. We believe our data falls into this category. 

Furthermore, below we list the criteria of performing this analysis and how our dataset, to our knowledge, satisfies the criteria: 

1. Assumption #1: The dependent variable should be measured at the ordinal level. 

In our study, our outcome/dependent variables were Knowledge, Attitudes and Practices tiered as Good, Medium or Poor based on Bloom’s cutoff points. 

2. Assumption #2: One or more independent variables that are continuous, ordinal or categorical (including dichotomous variables). 

3. Assumption #3: There is no multicollinearity. 

4. Assumption #4: You have proportional odds, which is a fundamental assumption of this type of ordinal regression model; that is, the type of ordinal regression that we are using in this guide (i.e., cumulative odds ordinal regression with proportional odds).

We have checked this using the Brant’s test for the three analyses, and this criterion was satisfied. 

According to a paper by Scott et al., 1997, they write that:

“Ordinal regression is a relatively new statistical method developed for analyzing ranked outcomes. In the past, ranked scales have often been analyzed without making full use of the ordinality of the data or, alternatively, by assigning arbitrary numerical scores to the ranks. While ordinal regression models are now available to make full use of ranked data, they are not used widely... We conclude that ordinal regression is a tool that is powerful, simple to use, and produces an interpretable parameter that summarizes the effect between groups over all levels of the outcome.”

Furthermore, it is our view that predictive models do not serve only to forecast, but in our case, this model is explaining the outcome, i.e. the factors associated with the KAP tiers.

This analysis, in our view, answers the three research questions:

1. what determines good Knowledge levels about COVID-19 in HCWs at Kiambu county, Kenya? 

2. what determines good attitude regarding COVID-19 in HCWs at Kiambu county, Kenya? 

3. what determines good practices towards COVID-19 in HCWs at Kiambu county, Kenya? 

We have thus used multivariable multilevel ordinal logistic regression to tease out which independent variables predict good KAP (separately) in Kiambu county, Kenya using a standard ordinal dependent outcome based on Bloom’s cutoff points. 

Thus, we have already applied statistical techniques that used two or more independent variables to predict the outcome of a dependent variable that is ordinal (multivariable multilevel ordinal logistic regression). 

Therefore, in our view, we do not think that multivariate analyses are appropriate for this study, given that multivariate indicates the simultaneous observation and analysis of more than one outcome variable. We do not view Knowledge, Attitude and Practice as one composite outcome, rather as three standalone outcomes. We also wish to state that our results were congruent between the initial and current analysis. 

Furthermore, other KAP papers published in this esteemed and respected journal have followed similar statistical methods to model ordinal outcomes (Balegha et al., 2021).

We humbly and kindly ask that you consider the above analysis for this manuscript. However, if this remains unsatisfactory, kindly provide additional clarification on this matter and afford us the chance to adjust our analyses. We thank you for your input regarding our manuscript. 

In addition, please find below our complete description of our statistical methods regarding this analysis, as it appears in the manuscript, for the construction of these multivariable ordinal logistic regression models and for accurate reproducibility: 

Multivariable Ordinal logistic regression models were constructed to determine the factors associated with good: (1) Knowledge, (2) Attitudes and (3) Practices of HCWs towards COVID-19. The models each assessed the associations of the independent variables with the KAP scores (dependent variables).

The link functions were chosen based on the changes in the cumulative probabilities: the “probit” link function for gradual changes (as noted with the knowledge and attitude types) and the complementary log-log link function for practice, for which the cumulative probabilities increased from 0 slowly and then rapidly approached 1 (Chapter 12 Ordinal Logistic Regression | Companion to BER 642: Advanced Regression Methods, n.d.).

The crude odds ratio (COR) was obtained for each independent variable per dependent variable. These values can be found in the S2 Table. Independent variables with a COR p-value of < 0.05 was used to construct the initial fitted model. Then, a null model was run for each dependent variable. The results of ordinal logistic regression are not valid unless they satisfy Brant’s test. To further validate the initial fitted models, each one underwent a check for satisfaction of the parallel regression/proportional odds assumption using the Brant test (Brant, 1990). This posits that the slopes (represented by β-coefficients/odds ratios) of the model across various ordinal outcome categories remain constant, while the intercepts can vary (Balegha et al., 2021). The Brant test serves to evaluate the overall significance of the model as well as the individual significance of all explanatory variables included in the model, where a significant test result indicates the model is invalid (Balegha et al., 2021).

The variables that did not satisfy Brant’s test were removed and the final fitted model was obtained. The results were finally tabulated as the β-coefficient, adjusted odds ratios (AOR) at 95% confidence intervals (CI), and the p-value. Further details regarding each model can be found below. 

KAP ordinal logistic regression modelling

For knowledge, the null model was fitted (Aikake Information Criterion, AIC = 903.2; Residual Deviance: 899.1). The independent variables that had a significant COR were: sex, education, cadre, news, international, social media, medical fora and journals, and these were used to construct the initial fitted model. This initial fitted model was checked using Brant’s test and predictor variables that failed the test were removed. As a result, cadre, news, international sites, social media and journals constituted the final model (AIC = 804.2; Residual Deviance: 788.2) which satisfied Brant’s test (X2 = 10.9, df = 6, p = 0.09). 

For attitude, the null (AIC = 661.8; Residual Deviance: 657.8) was fitted and the model was saturated. The statistically significant explanatory variables were: education, cadre, government, social media and continuous medical fora as sources of information. The Brant’s test indicated that education possessed a significant p-value. Thus, this predictor variable was dropped and final model fitted (AIC = 618.7; Residual Deviance: 604.7) which satisfied Brant’s test (X2 = 0.96, df = 5, p = 0.97). 

For practice, the null model was run (AIC = 691.9; Residual Deviance: 687.9). The predictor variables with significant COR included: education, cadre, government, news and social media as information sources, and these were fitted into the initial fitted model. The Brant’s test for the significant variables indicated that education was significant (p < 0.05). Thus, the final model was fitted (AIC = 631.7; Residual Deviance: 617.7), which satisfied Brant’s test (X2 = 4.63, df = 5, p = 0.46). 

3. Have the authors made all data underlying the findings in their manuscript fully available?

Reviewer #1: Yes

Reviewer #2: No

Reviewer #2, Thank you for your observation regarding the availability of the data underlying our manuscript's findings. Our data was included as Supplementary Data S1. 

Editor’s response: The reviewer’s comments have fully addressed in the manuscript text.

Author’s response: Nothing further to address

4. Is the manuscript presented in an intelligible fashion and written in standard English?

Reviewer #1: Yes

Reviewer #2: No

We took heed of Reviewer #2’s comment and have revised our manuscript for clarity and conciseness. 

Editor’s response: The reviewer’s comments have partly addressed in the manuscript text.

Author’s response: We further reviewed and revised our manuscript for clarity, correctness and accuracy. We have corrected any and all typographical errors. Thank you for this comment. 

5. Review Comments to the Author

Dear Reviewer #1, 

Thank you for your thoughtful and detailed feedback on our manuscript. We sincerely appreciate the time and effort you dedicated to reviewing our work. We have carefully considered each of your points, and your insights have been invaluable in enhancing the quality and clarity of our research.

Please find our Editor’s responses below.

Reviewer #1: 

Abstract:

• Their objective does not center on exploring the KAP. The term "evaluating" suggests an assessment of interventions, treatments, or comparisons between pre and post studies.

We acknowledge your concern about the term "evaluating" and have revised it to "assessing" to better align with our objective. Thank you for pointing this out.

Editor’s response: The reviewer’s comments have not yet fully addressed in the manuscript text.

Author’s response: We have further revised our objective in the abstract as “We aimed to describe the knowledge, attitude, and practices (KAP) of HCWs during the COVID-19 pandemic in Kiambu county, Kenya.”

• They utilized multivariate logistic regression to adjust for other variables and estimate the adjusted odds ratio (AOR) associated with a 95% confidence interval. It might be better to incorporate these details in the methods section.

We have since revised our data analysis plan and have included it in the abstract as “Ordinal logistic regression”.

Editor’s response: The manuscript still requires further revision to address the reviewer's comments adequately. The authors should consider reanalyzing the data using multivariate regression models.

Author’s response: We have indicated our position in regards to the analysis and direction to perform multivariate regression models at the beginning of this response letter. We reiterate our request for further clarification and we thank you for your input regarding our manuscript. 

• Instead of presenting p-values as "p-value <0.001," it would be clearer to state them as "p=xxxe-10." 

Your suggestion regarding the presentation of p-values has been duly noted and implemented. We now express them fully for better clarity.

Editor’s response: The reviewer’s comments have not yet fully addressed in the manuscript tables.

Author’s response: All p-values have been stated completely down to 4 decimal places, and where necessary using scientific notation both in the text and tables of the manuscript. 

• Could you provide information on the number of healthcare workers with good/bad practices?

We appreciate your observation regarding the need for information on healthcare workers with good/bad practices. We have revised the abstract to include this important detail.

Editor’s response: The reviewer’s comments have not yet fully addressed in the manuscript text.

Author’s response: We apologize for our omission of the Medium and Poor KAP results. We have included them in the abstract as below: 

“43.0% had good knowledge, 17.5% positive attitudes, and 68.4% good practice. 23.0% had medium knowledge, 35.6% medium attitude, 15.7% medium practice, 34.0% poor knowledge, 46.9% poor attitude, 15.9% poor practice.”

• In conclusion, the authors did not mention the associated factors that need improvement for raising awareness.

We apologize for the oversight regarding the associated factors for raising awareness. Your feedback has prompted us to revise our conclusion, incorporating recommendations in line with the study findings. 

Editor’s response: The reviewer’s comments have partly addressed in the manuscript text. 

Author’s response: We have revised our conclusion to mention the areas of improvement:

“This study emphasizes the substantial impact that governing bodies have on shaping favorable KAP. As a result, it's crucial for local government platforms to prioritize the dissemination of up-to-date information that aligns with international standards. This information should be tailored to the specific region, focusing on addressing deficiencies in healthcare practices and patient management. The identification of a significant number of HCWs lacking confidence in managing COVID-19 patients and feeling unprotected underscores a clear need for improvement in their understanding and implementation of preventive measures. This gap can be bridged by adequately equipping HCWs with locally manufactured PPEs. This aspect is crucial for pandemic preparedness, and we further advocate for the creation of a locally produced repository of medical equipment. These actions are pivotal in improving future crisis management capabilities.”

Methods:

• The authors briefly mention the study design, but a more detailed explanation is necessary.

Thank you for pointing out the need for a more detailed explanation of our study design. We have provided additional information to ensure a comprehensive understanding.

Editor’s response: The reviewer’s comments have fully addressed in the manuscript text.

Author’s response: Nothing to address

• The sample size calculation provided is quite general and lacks important information, such as the proportion (p) of healthcare workers in the study. A 50% proportion is not appropriate for this type of study. Based on previous research, the authors should calculate the sample size specifically for healthcare workers (HCWs).

• It is stated that this is an online survey, and the authors mention that the survey was conducted until they reached a sample size of 385 HCWs. However, it appears that this survey did not take the sample proportionally, and further clarification is needed.

Your concerns about the sample size calculation have been addressed. We have recalculated the sample size specifically for healthcare workers based on relevant literature and clarified our approach for achieving a representative sample, as below: 

We revised the sample size calculation based on a Ugandan study by Olum et al. (from which the questionnaire was adopted), where the proportion of sufficient KAP was 69%, 21% and 74%, respectively. We used these KAP scores to calculate the minimum required sample sizes for each outcome variable. We assumed a 95% confidence level (corresponding to a Z-score of 1.96) and a margin of error E of 5% (expressed as 0.05).

1. Knowledge: n = 323 HCWs.

2. Attitudes: n = 257 HCWs.

3. Practices: n = 307 HCWs.

To further enhance the representativeness of the sample, the weights of the specific cadres were targeted to mirror the population, thus, caregivers were to constitute 78% of the sample, administrative staff approximately 12.1%, and environmental health health 9.7% of the sample size. 

In our study, 438 HCWs responded. 

Editor’s response: The reviewer’s comments have fully addressed in the manuscript text.

Author’s response: Nothing further to address

Results:

• In lines 146-147, it would be helpful to provide the number of indicators and the total score of knowledge for better clarification.

We have revised the discussion to provide the number of indicators and the total score of knowledge for better clarification, as suggested.

Editor’s response: The reviewer’s comments have fully addressed in the manuscript text.

Author’s response: Nothing further to address

• In line 189, it is not possible for the 95% confidence interval of the odds ratio (OR) to be negative (-0.05 to 0.15).

Your observation about the negative 95% confidence interval of the odds ratio has been rectified. We have since revised our data analysis to ensure accurate reporting of the results and ensure none of the reported ORs have a negative sign. 

Editor’s response: The reviewer’s comments have fully addressed in the manuscript text.

Author’s response: Nothing further to address

• The study suggests an inverse association between practices and attitudes with knowledge. Is this interpretation accurate?

Regarding the interpretation of the inverse association between practices and attitudes with knowledge, we have reviewed the findings and confirmed the accuracy of our interpretation. This has been elaborated upon in the discussion section. 

Editor’s response: The reviewer’s comments have not yet fully addressed in the manuscript text. The authors should require reanalysis the data using statistical models and interpretation properly.

Author’s response: We wish to clarify this section once again. The reviewer’s comment alluded to the section of our manuscript where we carried out Spearman’s rank correlations between the Knowledge, Attitudes and Practice scores. This has been discussed in the Methods section under the heading DATA ANALYSIS, line 230 – 232, as below: 

“Correlations between scores of KAP were also analyzed using Spearman’s rank correlations given that the data were non-normally distributed. Correlations were interpreted using the following criteria:0–0.25 = weak correlation and 0.25–0.5 = fair correlation.”

From this analysis, our results were discussed under the heading Correlation between knowledge, attitude and practice scores in the Results section. 

There was a statistically significant fair correlation between knowledge and practice (r = -0.3300, p = 2.766×10-11), a weak correlation between knowledge and attitude (r = -0.1541, p = 6.538×10-3) and a fair correlation between attitude and practice toward COVID-19 disease (r = 0.3443, p = 6.843×10-10). These correlations can be found in Figure 3.

Discussion:

• In lines 241-244, the authors suggest that there were more gaps in knowledge and practices among HCWs. They found that HCWs did not exhibit appropriate practices, yet they had sufficient knowledge. However, their data analysis did not support this observation, as it showed that HCWs with insufficient knowledge had good practices. These results appear contradictory.

We appreciate your feedback on the potential contradiction in our interpretation of knowledge and practices among HCWs. We have revisited our discussion and made necessary adjustments to align with the study's results more accurately.

Editor’s response: The reviewer’s comments have not yet fully addressed in the manuscript text.

Author’s response: We had revised our discussed in line with our findings. For this particular finding, we have discussed it in the section of our Discussion titled “KAP relationship”, as below: 

“In our study, we observed an inverse relationship between the scores for knowledge, attitudes, and practices among HCWs. Surprisingly, higher knowledge scores were associated with less favourable attitudes and practices among HCWs. This inverse relationship was notably stronger between knowledge and practice. However, we found that a positive attitude was directly correlated with the adoption of correct practices. These observations align with findings reported in a study conducted in Ethiopia [7]. However, these relationships, particularly between knowledge and attitude and knowledge and practice, are in contrast with studies from Nepal and Uttar Pradesh [8,43]. This is a peculiar finding as convention dictates that the more one knows the better their practices and attitude. Therefore, in our context, regardless of acquiring knowledge about COVID-19, this did not further translate into optimal practice or a positive attitude. This contradicts evidence that efficacy beliefs predict behaviour, which has been shown for a myriad of different diseases [44–48]. Therefore, in addition to being knowledgeable, Kiambu County HCWs needed to be further convinced that the practices for its prevention would be effective.”

• In line 277, a major limitation of the study was its cross-sectional, which prevented the authors from conducting a more in-depth exploration of the KAP items.

Your point about the limitation of the study being cross-sectional has been duly noted and included in our limitations section.

Editor’s response: The reviewer’s comments have partly addressed in the manuscript text.

Author’s response: We had included this as a limitation, as below:

“This study has limitations, such as being cross-sectional, which prevented a more in-depth exploration of each KAP item. We also did not take into account the experience of HCWs in the private sector. The sampling strategy was also voluntary purposive sampling. Additional characteristics linked with COVID-19 behaviours such as perceived obstacles or other communication aspects were not evaluated in this study. Similar to other surveys, ours may also be affected by response bias. However, this study provided vital information on the KAP of HCWs in the Kenyan context.”

• While the authors presented a well-written conclusion, it would be beneficial to provide further clarification on the factors that need improvement and how to address them.

Thank you for highlighting the need for further clarification on the factors requiring improvement and how to address them. We have revisited our conclusions and provided additional details to address this concern. 

Once again, we are grateful for your thorough review and constructive feedback. Your input has significantly strengthened the quality and cohesiveness of our manuscript.

 

Dear Reviewer #2,

We appreciate your thoughtful feedback and valuable suggestions regarding our manuscript. Your insights have been extremely instrumental in improving the clarity and quality of our work. We have carefully considered each of your points and made the necessary revisions to address your concerns. We sincerely thank you for your time in reviewing this paper as it is our belief that your comments have greatly elevated the quality of our work. 

Please find our Editor’s responses below. 

Reviewer #2: 

Title

The title is not easy to understand for the readers

We understand your concern about the readability of the title. Based on your feedback, we have revised the title to make it more straightforward and accessible to readers: "Knowledge, Attitudes, and Practices towards COVID-19 among Healthcare Workers: A Cross-Sectional Survey from Kiambu County, Kenya." We hope this new title is more comprehensible.

Editor’s response: The reviewer’s comments have fully addressed in the manuscript text.

Author’s response: Nothing further to address

Abstract

• Background is well written

Thank you for this encouraging comment.

• The study period (May to August) is not consistent with the mentioned period in the study design on page 4. The questionnaire was used to collect data not for determining associated factors of KAP. Simple and multiple logistic regression, not multivariate logistic regression. See also comments in the methods and materials section. 

We appreciate your attention to detail regarding the study period and the use of the questionnaire. We have made the necessary adjustments to accurately reflect the study period in all instances as 11th March 2021 to 12th August 2021. Thank you for the astute correction regarding the questionnaire and data analysis, we have revised the statement to read “A structured questionnaire was used to explore the factors associated with COVID-19-related KAP”. We have since revised our approach to data analysis as ordinal logistic regression and we have indicated the same in the abstract. 

Editor’s response: The manuscript still requires further revision to address the reviewer's comments adequately. The authors should consider reanalyzing the data using multivariate regression models.

Author’s response: We have addressed the comment regarding your direction to perform multivariate analysis above. Kindly, we ask for further direction and clarification regarding this matter. 

• Results should be revised according to the comments raised in the methods and materials sections

We have revised the results section in alignment with the updated data and methodology.

• ‘Messenger’ word should be rephrased; authors used this term in the whole manuscript. 

Your feedback on the use of the term "messenger" has been duly noted, and we have rephrased it throughout the manuscript for consistency and clarity. 

Editor’s response: The reviewer’s comments have fully addressed in the manuscript text.

Author’s response: Nothing further to address

Keyword

• Most of the keywords are overlapped with the title of this study

We understand your concern about the overlap between keywords and the title. To address this, we have revised the keywords to ensure they are distinct from the title: Knowledge, attitudes, practices; Healthcare workers; pandemic; Health communication; efficacy beliefs. 

Editor’s response: The reviewer’s comments have fully addressed in the manuscript text.

Author’s response: Nothing further to address

Introduction

• The introduction section is not shaped and written as required for scientific research

We appreciate your feedback on the introduction section. We have restructured and rewritten the introduction to meet the scientific research standards.

Editor’s response: The reviewer’s comments have not yet fully addressed in the manuscript text.

Author’s response: We appreciate your insightful observation. By conducting an exhaustive review of pertinent background information on the KAP of HCWs regarding COVID-19, we have revised the introductions and honed in on our particular thesis statement. Additionally, we have modified the introduction to be more captivating to your readers.

•The authors did not consider quality points to highlight why they conducted this study, and about research gaps

The revised introduction now includes quality points highlighting the reasons for conducting the study. 

Editor’s response: The reviewer’s comments have fully addressed in the manuscript text.

Author’s response: Nothing further to address

•Need a comprehensive review of relevant literature

The revised introduction now includes a comprehensive review of relevant literature. Thank you. 

Editor’s response: The reviewer’s comments have fully addressed in the manuscript text.

Author’s response: Nothing further to address

• The objective of this study is not clearly mentioned.

We revised the objective of the study at the end of the Introduction section “This study sought to determine the KAP of HCW towards COVID-19 in Kiambu county, Kenya.”. 

Editor’s response: The reviewer’s comments have not fully addressed in the manuscript text.

Author’s response: Thank you for this comment. For additional clarity, we have rewritten the thesis statement at the end of the introduction section as follows: “Thus, the KAP of HCW towards COVID-19 were critical to the success of the overall COVID-19 response. To explore this hypothesis, we carried out a cross-sectional study aimed at describing HCWs' KAP regarding COVID-19 in Kiambu County, Kenya..”

•What are the implications of the findings of this study

The revised introduction now includes a paragraph in the introduction section to cover this, as below: 

“However, given the paramount role of HCWs in pandemic response, what they know, feel and do becomes crucial in the face of a public health emergency. This situation is further complicated by the “infodemic” that paralleled the COVID-19 pandemic (Singh et al., 2022), and by a supply chain crisis that affected the availability of PPEs for adhering to proper protocols (Shah et al., 2021). To avert the higher rate of being infected among HCWs, equipping them with good knowledge and practice is imperative, especially in countries with already low health worker-to-population ratio (Fetansa et al., 2021). Thus, the KAP of HCW towards COVID-19 were critical to the success of the overall COVID-19 Editor’s response.”

Methods and Materials

• The definition of the study population is not clear. It is also not clear, who are actually HCWs in this study or the way authors defined HCW, required quality modification. 

The above two observations have been noted, and thus the definitions have been revised according to the categories used in our study. In this study, a HCW included any person involved in the provision of health services to a user or those who are on facility grounds employed by the facility. At the time of the study, the county of Kiambu was served by a total of 3700 HCWs. These HCWs were divided into caregivers, administrative staff and environmental health workers, and details of each category can be found below. 

1. Caregivers included all HCWs who interact with patients directly. They included: medical officers, consultants, nurses, clinical officers, dental officers, dental technologists, pharmacists, pharmaceutical staff, laboratory staff, orthopedic technologist, nutritionists, radiographers, physiotherapists and mortuary attendants. 

2. Administrative staff included HCWs who do not interact with patients directly. They included: health administrative officers and staff, health-supportive staff, medical engineering technologists, health records & information officers, medical social workers, ambulance drivers, and HIV testing services staff.

3. Environmental health workers consisted of public health staff, such as community health volunteers, health promotion officers, and public health officers/community health officers

Editor’s response: The reviewer’s comments have not fully addressed in the manuscript text.

Author’s response: In our study tool, each respondent was allowed to state their cadre from a range of options as below: 

medical officers, consultants, nurses, clinical officers, dental officers, dental technologists, pharmacists, pharmaceutical staff, laboratory staff, orthopedic technologist, nutritionists, radiographers, physiotherapists and mortuary attendants, health administrative officers and staff, health-supportive staff, medical engineering technologists, health records & information officers, medical social workers, ambulance drivers, and HIV testing services staff, community health volunteers, health promotion officers, and public health officers/community health officers. 

Based on their responses, during analysis they were sorted and grouped into the three broad categories used in the manuscript: 1) Caregivers, 2) Administrative staff and 3) Environmental health workers. 

It's important to note that these categorizations align with the cadre groupings established by the Government of Kenya (GOK), defining the specific healthcare worker population under study. We trust that this clarification adequately addresses the categorization process used in our research.

• According to the nature of the study setting and study population, the sample size determination and sampling technique are inappropriate. The sampling on a volunteer basis is not scientifically robust.

In this study, we employed voluntary purposive sampling. While this approach would not be the best, we had to specifically target HCWs who then had to consent to participate in the study. This approach has been used previously in similar KAP studies, such as one by Mark et al., 2022, and was adopted here. 

Editor’s response: The reviewer’s comments have fully addressed in the manuscript text.

Author’s response: Nothing further to address

• It is confusing, what type of questionnaire the authors used to collect the data: structured/open-ended/semi-structured? How could a questionnaire be structured and open-ended simultaneously? Probably, the authors considered a semi-structured research tool

Thank you for this observation. It is correct that we employed a semi-structured questionnaire to collect data, but this manuscript only details the results of the structured questionnaire. We have corrected the manuscript accordingly. 

Editor’s response: The reviewer’s comments have fully addressed in the manuscript text.

Author’s response: Nothing further to address

• The authors did not mention how they checked the reliability, validity, and consistency of their data and questionnaire

This is a good observation and has been addressed in the manuscript in the section headed as “Questionnaire”. The main additions can be found below:

We employed a questionnaire from Olum et al.(Olum et al., 2020) for this study. This questionnaire was constructed as such: 

1. Knowledge was assessed using a 11-item questionnaire adapted from Zhong et al., and modified to suit HCWs, each correct answer weighing one point. According to Zhong et al., this was a reliable knowledge scale for adoption as it possessed a Cronbach’s alpha score of 0.71 (Zhong et al., 2020). 

 We adopted this Knowledge scale for our study, however we added three context-specific questions regarding the location of Kiambu county’s COVID-19 isolaton centre, handling a suspected COVID-19 case and data entry for suspected COVID-19 cases. These questions were selected and reviewed by public health experts. 

2. Attitudes were assessed using 5 Likert-item questions that have been adopted from Goni et al. and modified by Olum et al. Goni et al.’s Cronbach’s alpha scale measurement for this was 0.77 (Goni et al., 2020).

We adopted this Attitude scale for our study, however we added six context-specific questions regarding how the national and county government structures were handling the pandemic at various stages.

3. Practices were assessed using five Likert-item questions that have been developed from the WHO recommended practices. This was adopted fully from the Olum et al.

The alpha measurements for the scale subsets as used in our study were as follows: αk = 0.69, αa = 0.81 and αp = 0.61. In this study, a Cronbach’s alpha score of above 0.6 was considered adequate, similar to other KAP studies (Gopalakrishnan et al., 2021; Majmundar et al., 2018; Mark et al., 2022; Sayili et al., 2022). 

Editor’s response: The reviewer’s comments have fully addressed in the manuscript text.

Author’s response: Nothing further to address

• The authors did not mention what was the basis of the classification of the target variables

This is duly noted. We have revised our study to use Bloom’s cutoff points as the basis of classifying our target variables into Good (>80%), Medium (60-79%) and Poor (<60%). 

Editor’s response: The reviewer’s comments have fully addressed in the manuscript text.

Author’s response: Nothing further to address

• The authors did not check the bivariate association of the target variables with the predictor variables, so, how did they select probable predictor variables for multiple regression modeling? In the modeling, some potential predictors may be insignificant due to arbitrary selection of the predictors. 

Given this comment, we had revisited the analysis and revised the data analysis plan to conduct ordinal logistic regression given the application of Bloom’s cutoff points that divided each KAP target variable into Good, Medium and Poor. 

Editor’s response: The reviewer’s comments have fully addressed in the manuscript text.

Author’s response: Nothing further to address

• The authors actually performed binary and multiple logistic regression but they mentioned “multivariate logistic regression”

This is duly noted and has been modified according to our new analysis. Thank you. 

Editor’s response: The reviewer’s comments have not yet fully addressed in the manuscript text.

Author’s response: Dear Editor, we refer you to our comment at the beginning of this response letter to clarify our stance on the analysis used in this manuscript. We, again, ask for your clarification based on our view that our current analysis satisfies the criterion for the performance of multivariable ordinal logistic regression. 

• How the authors selected the link function for logistic regression as they did not check the nature of the target variable

Thank you for this observation. We have revisited our analyses and reorganized our target variable according to the Bloom’s cutoff points for KAP studies and have organized our outcomes, such as type of knowledge into: Good, Medium and Poor. 

The link function was chosen based on the changes in the cumulative probabilities: the “probit” link function for gradual changes (as noted with the knowledge and attitude types) and the complementary log-log link function for practice, for which the cumulative probabilities increased from 0 slowly and then rapidly approached (Chapter 12 Ordinal Logistic Regression | Companion to BER 642: Advanced Regression Methods, n.d.).

Editor’s response: The reviewer’s comments have not yet fully addressed in the manuscript text.

Author’s response: Given that we revised our analysis to multivariable ordinal logistic regression, we thus adjusted our link functions to allow for the explanation of the ordinal outcome of knowledge, attitudes and practices. 

Again, we request clarification on this matter. 

Results

• The results should be revised according to the comments in the methodology section

This was duly noted and the entirety of the results section has been revised thoroughly. 

 Editor’s response: The reviewer’s comments have fully addressed in the manuscript text.

Author’s response: Nothing further to address

Authors should also consider the following specific observations:

• The results are poorly written and presented

Thank you for this observation. We have rewritten the entirety of the results section again for enhanced clarity. 

 - We have thoroughly reviewed and refined the presentation of our results. The results of the ordinal logistic regression now include all the KAP outcomes and only the significant variables. This approach enhances the clarity of our findings and makes the presentation more precise and accessible to readers.

We hope these revisions address your concerns about the clarity and presentation of our results. 

Editor’s response: The reviewer’s comments have partly addressed in the manuscript text.

Author’s response: We have further reviewed and revised the Results section for added clarity and presentation of our results. We have additionally improved the presentation of the questions in our tables to be more succinct and to the point. 

• Tables are too long

Thank you for your observations regarding the presentation of the results. We greatly value your feedback and have taken your comments into serious consideration. Based on your suggestions, we have made the following revisions to address your concerns:

1. Tables Length:

 - We acknowledge (and share) your concern about the length of the tables. To address this, we have revised Table 1 to include only the socio-demographic characteristics of the sample. Cadre proportions have been mentioned in the narrative section of the results, ensuring a more concise presentation of information. We have also included the data pertaining to the specific cadres as Supplementary material S3. 

 - Additionally, we have truncated the other tables where possible, ensuring a more focused and streamlined presentation of the data.

- We have also included the crude odds ratio results and the p-values as Supplementary material S2. 

Editor’s response: The reviewer’s comments have not fully addressed in the manuscript text. Instead the authors may create graphs for presenting the data. 

Author’s response: We added figures to denote the proportions of Good, Medium and Poor KAP for all three outcomes (Figure 1). We also added a graph to illustrate the responses regarding the main clinical symptoms of COVID-19 (figure 2) and Figure 3 demonstrate the relationships between the KAP scores and the results of the linear correlation. We hope that these changes suffice and reduce the reliance on the tables. 

• Authors unnecessarily mentioned too long questions in the text

We have revised the results section to eliminate references to long questions, ensuring a more concise and reader-friendly presentation of our findings.

Editor’s response: The reviewer’s comments have partly addressed in the manuscript text.

Author’s response: We have taken the initiative to eliminate all references to lengthy questions within the manuscript text. Additionally, we have shortened the questions in the tables to maintain brevity and ensure their direct relevance to the content.

• Authors used ‘n’ for indicating sample size in the methodology section; in the result section it is confusing about the notation ‘n’ and ‘N’

We have rectified the use of 'n' and 'N' to accurately represent whether it refers to the sample or population size, ensuring clarity in both the methodology and results sections.

Editor’s response: The reviewer’s comments have fully addressed in the manuscript text.

Author’s response: Nothing further to address

• In Tables 4, 6, and 8, the authors provided a column ‘p-value’. P-value of which estimates: crude odds ratio (COR) or adjusted odds ratio (AOR)? If it indicates AOR, then where is the significance value of the COR estimates? Without significance value, how these results could be interpretable?

We thank you again, for this invaluable comment and apologize for its omission. In the revised manuscript, we have included the coefficient, AOR (with confidence intervals in brackets) and p-values only.

Editor’s response: The reviewer’s comments have not yet fully addressed in the manuscript text.

Author’s response: We thank you again, for this invaluable comment and apologize for its omission.

In the revised manuscript, we have included the coefficient, AOR (with confidence intervals in brackets) and p-values only. The crude odds ratios and respective p-values can be found in the Supplementary material S2 due to the length of the tables. However, an explanation of the significant variables that made it to the final model can be found in the methods section, as below:

“For knowledge types, the null model was fitted (Aikake Information Criterion, AIC = 903.2; Residual Deviance: 899.1). The independent variables that had a significant COR were: sex, education, cadre, news, international, social media, medical fora and journals, and these were used to construct the initial fitted model. This initial fitted model was checked using Brant’s test and predictor variables that failed the test were removed. As a result, cadre, news, international sites, social media and journals constituted the final model (AIC = 804.2; Residual Deviance: 788.2) which satisfied Brant’s test (X2 = 10.9, df = 6, p = 0.09). 

For attitude types, the null (AIC = 661.8; Residual Deviance: 657.8) was fitted and the model was saturated. The statistically significant explanatory variables were: education, cadre, government, social media and continuous medical fora as sources of information. The Brant’s test indicated that education possessed a significant p-value. Thus, this predictor variable was dropped and final model fitted (AIC = 618.7; Residual Deviance: 604.7) which satisfied Brant’s test (X2 = 0.96, df = 5, p = 0.97). 

For practice types, the null model was run (AIC = 691.9; Residual Deviance: 687.9). The predictor variables with significant COR included: education, cadre, government, news and social media as information sources, and these were fitted into the initial fitted model. The Brant’s test for the significant variables indicated that education was significant (p < 0.05). Thus, the final model was fitted (AIC = 631.7; Residual Deviance: 617.7), which satisfied Brant’s test (X2 = 4.63, df = 5, p = 0.46).”

• It is clear from Tables 4, 6, and 8, that the authors performed simple logistic regression for all predictor variables as well as performed multiple logistic regression including all the predictor variables in the model, Why? Why did the authors consider the predictors with insignificant COR in the multiple logistic regression modeling?

We thank you again, for this invaluable comment. We have revised the manuscript according to the refined results of ordinal logistic regression. For this analysis, we only included predictors with COR with p-values less than < 0.05 in our initial and final fitted models. 

Editor’s response: The reviewer’s comments have not fully addressed in the manuscript text. The authors should consider multivariate associated models instead of predictive models.

Author’s response: We would like to ask for further clarification for the same before revising our analysis protocol, as stated above. Thank you. 

• Estimates and p-values are not presented in a unique and standard format.

The estimates and p-values in our results section have been restructured to adhere to a standard format, enhancing the consistency and readability of the presentation.

Editor’s response: The reviewer’s comments have fully addressed in the manuscript text.

Author’s response: Nothing further to address

• The second column of Tables 4, 6, and 8 should be renamed as ‘Label’

Due to the restructuring of our analysis, the specific tables you referred to no longer exist. Nevertheless, we appreciate your suggestion, which has guided our approach in creating clearer and more organized tables. We are grateful for the direction. 

Editor’s response: The reviewer’s comments have fully addressed in the manuscript text.

Author’s response: Nothing further to address

• In Table 4, the authors provided COR for attitude and practice considering these as predictor variables of knowledge. In principle, how attitude and practice could be the predictors of knowledge? Is there any basis or reference against this issue? If yes, why there is no adjusted odds ratio for these factors? Probably, the authors considered it wrongly. Authors should rethink and redesign

We have since rethought and redesigned our analysis of our data to ordinal logistic regression. We have only considered the independent variables as predictors for each KAP dependent variable. These variables were chosen only if they had a crude odds ratio COR of less than p < 0.05. 

Editor’s response: same as before. The author should require reanalysis data using associated regression models instead of predictive models. 

Author’s response: We wish to reiterate that multivariable ordinal logistic regression is a model that uses one or more explanatory variables to explain an ordered dependent outcome (such as good, medium and poor KAP). 

Thus, in our view, we have already applied a regression model to explain the outcomes of the dependent variables. 

• In Tables 6 and 8, the authors did not consider the knowledge level as a predictor variable for attitude and practice, why?

We have since rethought and redesigned our analysis of our data to ordinal logistic regression. This approach ensures a more accurate representation of the relationship between predictor variables and outcomes, addressing the concerns you raised regarding the inclusion of attitude, practice, and knowledge levels as predictors.

Editor’s response: The reviewer’s comments have fully addressed in the manuscript text.

Author’s response: Nothing further to address

• Is the result of correlation measurement between knowledge and practice correct? If yes, does this type of relationship exist in reality? The authors did not discuss the reason for this result in the discussion section. What would be the implications of this finding?

In Editor’s response to your query, we did conduct Spearman’s rank correlation for the non-normally distributed knowledge scores against the practice scores. The analysis indeed revealed an inverse relationship between knowledge and practice, which was statistically significant. However, we acknowledge that our discussion of this relationship in the manuscript was not as clear and comprehensive as it should have been.

Your feedback has prompted us to reevaluate our approach, and we have decided to restructure the discussion section into distinct segments. Specifically, we will now have separate sections dedicated to discussing the knowledge, attitude, and practice findings individually. Additionally, we will include a dedicated section to explore the relationship between knowledge and practice in greater depth. This restructuring will enable us to thoroughly analyze the implications of this finding and provide a more detailed and nuanced discussion on this significant aspect of our research.

We are genuinely grateful for your astute observation, which has guided us to enhance the clarity and depth of our discussion. 

Editor’s response: The reviewer’s comments have fully addressed in the manuscript text.

Author’s response: Nothing further to address

Discussion

• The discussion section is not written as expected for a scientific article

In Editor’s response to your comments, we have diligently revised the discussion section to ensure a proper and accurate reflection of our methods and results.

Editor’s response: The reviewer’s comments have partly addressed in the manuscript text.

Author’s response: We have taken the initiative to revise and restructure our discussion section in order to comprehensively address each pertinent finding from our study, with the aim of ensuring that every finding aligns precisely with the results section and that our interpretations are accurately presented.

• The authors are suggested to revise the methodology and results according to the comments raised in the respective sections

We have taken time to review both the methods and results section for congruency with our discussion. 

Editor’s response: The reviewer’s comments have fully addressed in the manuscript text.

Author’s response: Nothing further to address

• They should rewrite this section by explaining the potential findings with proper supporting citations

We have taken care to address the concerns raised in the respective sections and have provided a comprehensive explanation of our study findings. Additionally, we have strengthened our discussion by incorporating relevant literature to support our conclusions. 

Editor’s response: The reviewer’s comments have fully addressed in the manuscript text.

Author’s response: Nothing further to address

• Authors are also suggested to add implications of the findings and limitation of the study

To further enhance the quality of our manuscript, we have included a dedicated section discussing the implications of our study findings. Furthermore, we have outlined the limitations of our study, acknowledging the constraints and challenges we encountered during our research.

Editor’s response: The reviewer’s comments have fully addressed in the manuscript text.

Author’s response: Nothing further to address

 

References quoted: 

Balegha, A. N., Yidana, A., & Abiiro, G. A. (2021). Knowledge, attitude and practice of hepatitis B infection prevention among nursing students in the Upper West Region of Ghana: A cross-sectional study. PLOS ONE, 16(10), e0258757. https://doi.org/10.1371/JOURNAL.PONE.0258757

Brant, R. (1990). Assessing proportionality in the proportional odds model for ordinal logistic regression. Biometrics, 46(4), 1171–1178. https://doi.org/10.2307/2532457

Chapter 12 Ordinal Logistic Regression | Companion to BER 642: Advanced Regression Methods. (n.d.). Retrieved November 4, 2023, from https://bookdown.org/chua/ber642_advanced_regression/ordinal-logistic-regression.html

Fetansa, G., Etana, B., Tolossa, T., Garuma, M., Tesfaye Bekuma, T., Wakuma, B., Etafa, W., Fekadu, G., & Mosisa, A. (2021). Knowledge, attitude, and practice of health professionals in Ethiopia toward COVID-19 prevention at early phase. SAGE Open Medicine, 9, 205031212110122. https://doi.org/10.1177/20503121211012220

Goni, M. D., Naing, N. N., Hasan, H., Wan-Arfah, N., Deris, Z. Z., Arifin, W. N., Hussin, T. M. A. R., Abdulrahman, A. S., Baaba, A. A., & Arshad, M. R. (2020). Development and validation of knowledge, attitude and practice questionnaire for prevention of respiratory tract infections among Malaysian Hajj pilgrims. BMC Public Health, 20(1), 1–10. https://doi.org/10.1186/S12889-020-8269-9/TABLES/5

Gopalakrishnan, S., Kandasamy, S., Abraham, B., Senthilkumar, M., & Almohammed, O. A. (2021). Knowledge, Attitude, and Practices Associated With COVID-19 Among Healthcare Workers in Hospitals: A Cross-Sectional Study in India. Frontiers in Public Health, 9, 787845. https://doi.org/10.3389/FPUBH.2021.787845

Majmundar, A., Allem, J. P., Cruz, T. B., & Unger, J. B. (2018). The Why We Retweet scale. PLoS ONE, 13(10). https://doi.org/10.1371/journal.pone.0206076

Mark, E., Udod, G., Skinner, J., & Jones, M. (2022a). Knowledge, attitudes, and practices [KAP] toward COVID-19: A cross-sectional study in the New York Metropolitan Area and California Bay Area. PLOS ONE, 17(8), e0271212. https://doi.org/10.1371/JOURNAL.PONE.0271212

Mark, E., Udod, G., Skinner, J., & Jones, M. (2022b). Knowledge, attitudes, and practices [KAP] toward COVID-19: A cross-sectional study in the New York Metropolitan Area and California Bay Area. PLoS ONE, 17(8 August). https://doi.org/10.1371/journal.pone.0271212

Olum, R., Chekwech, G., Wekha, G., Nassozi, D. R., & Bongomin, F. (2020). Coronavirus Disease-2019: Knowledge, Attitude, and Practices of Health Care Workers at Makerere University Teaching Hospitals, Uganda. Frontiers in Public Health, 8. https://doi.org/10.3389/fpubh.2020.00181

Sayili, U., Siddikoglu, E., Pirdal, B. Z., Uygur, A., Toplu, F. S., & Can, G. (2022). The heat wave knowledge, awareness, practice and behavior scale: Scale development, validation and reliability. PLoS ONE, 17(12 December). https://doi.org/10.1371/journal.pone.0279259

Scott, S. C., Goldberg, M. S., & Mayo, N. E. (1997). Statistical assessment of ordinal outcomes in comparative studies. Journal of Clinical Epidemiology, 50(1), 45–55. https://doi.org/10.1016/S0895-4356(96)00312-5

Shah, W., Heightman, M., & O’brien, S. (2021). UK guidelines for managing long-term effects of COVID-19 Effective supply chain surveillance for PPE. The Lancet, 397, 1706–1707. https://doi.org/10.1007/s11606¬020¬06230¬1

Singh, K., Lima, G., Cha, M., Cha, C., Kulshrestha, J., Ahn, Y. Y., & Varol, O. (2022). Misinformation, believability, and vaccine acceptance over 40 countries: Takeaways from the initial phase of the COVID-19 infodemic. PLoS ONE, 17(2 February). https://doi.org/10.1371/journal.pone.0263381

Zhong, B. L., Luo, W., Li, H. M., Zhang, Q. Q., Liu, X. G., Li, W. T., & Li, Y. (2020). Knowledge, attitudes, and practices towards COVID-19 among Chinese residents during the rapid rise period of the COVID-19 outbreak: a quick online cross-sectional survey. International Journal of Biological Sciences, 16(10), 1745. https://doi.org/10.7150/IJBS.45221

---

## [Editor Report · Decision Letter 2]

3 Jan 2024

Knowledge, attitudes and practices towards COVID-19 among healthcare workers: a cross-sectional survey from Kiambu County, Kenya

PONE-D-23-19304R2

Dear Dr. Juttla,

We’re pleased to inform you that your manuscript has been judged scientifically suitable for publication and will be formally accepted for publication once it meets all outstanding technical requirements.

Kind regards,

Probir Kumar Ghosh, MSc.

Guest Editor

PLOS ONE
---

## [Editor Report · Acceptance letter]

2 Mar 2024

PONE-D-23-19304R2 

PLOS ONE

Dear Dr. Juttla, 

I'm pleased to inform you that your manuscript has been deemed suitable for publication in PLOS ONE. Congratulations! Your manuscript is now being handed over to our production team.

Kind regards, 

on behalf of

Mr. Probir Kumar Ghosh 

Guest Editor

PLOS ONE